# Geospatial distribution and multilevel determinants of inadequate minimum dietary diversity and its consequences for children aged 6–23 months in Sub-Saharan Africa

Bayuh Asmamaw Hailu[1,2], Chala Daba[3,4], Yawkal Tsega[5], Ashebir Asaminew[6], Birhan Asmame Miheretu[7], Abel Endawkie[8]*

1 Monitoring and Evaluation, Wollo University, Dessie, Ethiopia, 2 Research Center for Inclusive Development in Africa (RIDA), Addis Ababa, Ethiopia, 3 Department of Environmental Health, College of Medicine and Health Sciences, Wollo University, Dessie, Ethiopia, 4 National Centre for Epidemiology and Population Health, The Australian National University, Canberra, ACT, Australia, 5 Department of Health System and Management, School of Public Health, College of Medicine and Health Sciences, Wollo University, Dessie, Ethiopia, 6 Development Bank, Addis Ababa, Ethiopia, 7 Department of Geography and Environmental Studies, Wollo University, Dessie, Ethiopia, 8 Department of Epidemiology and Biostatistics, School of Public Health, College of Medicine and Health Sciences, Wollo University, Dessie, Ethiopia

* abelendawkie@gmail.com

## Abstract

### Background

Inadequate minimum dietary diversity (MMD) is the leading cause of malnutrition among young children in Sub-Saharan Africa (SSA). The evidence of geospatial distribution and multilevel determinants of inadequate MDD and its consequence among children is important for the Sustainable Development Goal (SDG0) 2030 agenda. Therefore, this study aimed to determine the geospatial distribution and multilevel determinants of inadequate MDD and its consequences among children in SSA.

### Method

The study utilized recent Demographic and Health Surveys data including 57,912 children. Spatial and multilevel analyses were employed, and variables significantly associated with inadequate MDD and undernutrition with MDD consumption were assessed and significance was declared using a p-value threshold of <0.05. Adjusted odds ratio (AOR) with 95% confidence interval (CI) was reported.

### Results

The prevalence of inadequate MDD was 80.3% with distinct spatial variation. Spatial distribution showed that; Gabon, Cameron, Ethiopia, Democratic Republic of Congo, Chad, Mali, Burkina Faso, Ivory Coast, Liberia, and Senegal had a very high burden of inadequate MDD. Factors like children's age, maternal age, educational status,

**Data availability statement:** The data for this study was sourced from the DHS Program. Interested researchers can request access to the datasets after creating an account and submitting a request. Detailed access information is available on the DHS Program website (https://dhsprogram.com/data/Access-Instructions.cfm). The authors confirm that all interested researchers and third parties can access the data in the same manner as the authors, and that no special access privileges were granted to the authors.

**Funding:** The author(s) received no specific funding for this work.

**Competing interests:** The authors have declared that no competing interests exist.

**Abbreviation:** AIC, Akakian information Criteria; AOR, Adjusted odds ratio; ANC, Antenatal care; BIC, Bayesian information Criteria; COR, crud odds ratio; EA, Enumeration Area;ICC, Intra-Class Correlation; MDD, Minimum Dietary Diversity; MEDHS, Mini Ethiopian Demographic and Health Survey; MOR, Median Odds Ratio; PCV, Proportional Changes of Variance; SNNP, South Nation Nationalities and Peoples; SSA, Su-Saharan Africa; WHO, World Health Organization.

antenatal care (ANC)/ postnatal care (PNC) visits, no media exposure, wealth status, maternal stunting and wasting, and distance from health facilities were associated with inadequate MDD in SSA. The risk of anemia, stunting, and wasting were significantly associated with inadequate MDD among children in SSA.

## Conclusion

The prevalence of inadequate MDD in SSA is high. Spatial distribution revealed that inadequate MDD was prevalent in most areas of the Western, Northern, Eastern, and Central parts of SSA. Maternal and children's age, educational status, ANC/ PNC visits, no media exposure, wealth status, maternal stunting and wasting, and distance from health facilities were determinants of inadequate MDD in SSA. The spatial clustering of inadequate MDD in certain regions of SSA, suggests the need for geographically targeted interventions to address the determinants of inadequate MDD in these high-burden areas. The study revealed strategies should focus on promoting frequent ANC/ PNC visits, improving maternal nutrition, reducing poverty, and improving maternal employment status to reduce inadequate MDD among children. This study highlights a significant association between MDD and anemia, stunting, and wasting in children aged 6-–23 months. To address these critical issues, it is essential to improve MDD among children, as this intervention can play a vital role in achieving SDG target 2.2, which aims to end all forms of malnutrition by 2030.

## Introduction

The World Health Organization (WHO) and the United Nations Children's Fund (UNICEF) define Minimum Dietary Diversity (MDD) as a child who consumes foods from at least five out of eight designated food groups. These eight groups include: breast milk, grains, roots and tubers; legumes and nuts; dairy products; flesh foods (such as meat, fish, poultry, and organ meats); eggs; vitamin A-rich fruits and vegetables; and other fruits and vegetables [1].

Between the ages of 6 and 23 months, children's nutrient requirements per body weight rise, leading to breast milk being inadequate to fulfill all their needs [2]. Inadequate dietary diversity is a primary cause of childhood malnutrition. The malnutrition results in the most serious roadblocks to long-term socioeconomic growth and poverty reduction in the world [3, 4]. Malnutrition in early childhood is associated with inadequate diet diversity [5–7].

Globally, from 151 million children under the age of five, two billion children had inadequate dietary diversity and were stunted [8]. It is most common in low and middle-income countries, where it has a significant impact on increasing mortality and disease distribution [9–11]. Particularly in Sub-Saharan Africa (SSA), the occurrence of insufficient MDD results in malnutrition such as anemia, stunting, and wasting, remains pronounced [12]. The Sustainable Development Goals (SDGs)

aim to address these challenges by 2030 [13]. Specifically, SDG 2 focuses on achieving food security and improving nutrition through universal access to safe and nutritious food to achieve SDG target 2.2 to end all forms of malnutrition [14]. However, in SSA, the situation remains contrary to these goals, with statistics showing a rise in malnutrition [15]. Despite the recommendation and significant advances in child nutrition, the prevalence of inadequate MDD is significantly high in SSA and ranges from 67–94 in SSA [16–19].

Different literatures showed that the women's education [19–21], the husband's educational status [19–22], child age [20,23], mass media exposure [19–21], maternal age [23, 24], occupational status [25, 26], wealth index [19–21], antenatal care [22,27], and residence [26] in were found to be associated with MDD among children aged 6–23 months in different countries of SSA. Analyzing data on Minimum Dietary Diversity (MDD) at regional, national, and continental levels can yield valuable insights for developing contextually relevant strategies and policies. Understanding the geospatial distribution of MDD both within and between countries, along with the multifaceted factors contributing to inadequate MDD and its effects on children aged 6–23 months, is crucial for achieving Sustainable Development Goal 2.2 by 2030, which aims to eliminate all forms of malnutrition. This analysis provides location-specific evidence that facilitates national and sub-national comparisons, tracks changes over time, and identifies high-risk populations. It also informs targeted interventions, guides policy decisions, and helps monitor progress toward nutrition objectives established by the WHO and SDGs in Africa.

However, despite the author's research efforts, evidence regarding the geospatial distribution of MDD and the various factors contributing to its inadequacy among children aged 6–23 months in Sub-Saharan Africa remains limited [28–30]. Therefore, this study aimed to identify the geospatial distribution and multilevel factors of inadequate MDD and its consequence among children aged 6–23 months in SSA using recent DHS data.

## Methods

### Study setting

The research was conducted in Sub-Saharan Africa, which is situated on the African continent and boasts a diverse population. Sub-Saharan Africa encompasses the regions lying south of the Sahara Desert, including Central Africa, East Africa, Southern Africa, and West Africa. This vast region comprises over 40 countries, with a population of approximately 1 billion people, and it is characterized by a rich and varied cultural heritage. The study utilized the most recent publicly available and nationally representative cross-sectional Demographic and Health Surveys (DHS) data from 33 Sub-Saharan African countries available from 2010–2020 (the detailed information on the specific DHS survey years and countries included are provided in (S1_Table)).

### Study design

A cross-sectional study design based on secondary data from the recent DHS in sub-Saharan Africa was conducted.

### Source population

The study population consisted of children aged 6–23 months from the selected Enumeration Areas (EAs) in Sub-Saharan Africa (SSA).

### Data source

In the current study, we used the kids' records (KR) data set, and the dependent and independent variables were extracted at https://dhsprogram.com/ by contacting them through personal accounts after justifying the reason for requesting the data [31]. The data from each country, including individual-level information and geographic information system (GIS) coordinates, were consolidated into a single dataset. The data collected from the DHS survey was organized in a hierarchical structure, with households within a cluster forming the top level [32, 33].

 

## Sample size and sampling method

The study comprises 57,912 children (aged 6–23 months) from 33 SSA countries. Participants were selected using a two-stage stratified sampling technique. In the first stage, enumeration areas (EAs) were randomly chosen, followed by a random selection of households in the second stage.

## Variable measurement

The dependent variable was defined as follows: a child who consumed at least five of the eight food groups in the 24 hours prior to the interview was categorized as having adequate Minimum Dietary Diversity (MDD) and assigned a value of "0," while those who did not consume were classified as having inadequate MDD and assigned a value of "1" [1].

The household level factors like; child age, mother's age, media exposure, Antenatal Care (ANC) and Postnatal Care (PNC), maternal working status, maternal and partner education level, and maternal stunting and wasting; and in the community level factors like household wealth, ecology and distance to health facility were analyzed as independent variable for MDD consumption among children. In addition, we examined the impact of insufficient MDD among children. In this case, the dependent variable was the occurrence of malnutrition (such as anemia, stunting, and wasting,), while inadequate MDD served as one of the independent variables, which may lead to the occurrence of malnutrition among children.

## Data processing and analysis

Data cleaning and analysis were performed using STATA version 17.0. Descriptive statistics were utilized to characterize the study participants based on their socio-demographic characteristics, which were presented in text tables and figures. To account for the unequal probability of selection across geographically defined strata and non-responses, sample weights were applied.

## Spatial analysis

To conduct the spatial analysis, the weighted proportions of the outcome variable (MDD) by cluster number were calculated using STATA version 17. The spatial analysis itself was performed with QGIS version 3.28.15, SAT Scan version 9.6, and SAGA GIS version 2.3.2 statistical software.

## Mapping hotspot analysis

The hot spot analysis was conducted using the Getis-Ord Gi statistic* GeoDa version 1.14 which is part of a local hot spot analysis that helps to identify spatial clusters on the landscape. Either it pinpoints areas with statistically significant clustering of high values (hot spots) or low values (cold spots). Essentially, it tells us whether there is a meaningful pattern of clustering in the data. The null hypothesis assumes that there is no difference in characteristics between a given unit and its neighboring units. In other words, it checks if the observed clustering is more than what we would expect by random chance [34–37].

## Spatial interpolation

To estimate the prevalence in areas where data is not directly measured, we employed ordinary kriging within SAGA GIS (version 2.3.2) for spatial interpolation. This technique allows us to predict values at unmeasured locations based on the observed data from neighboring areas [38].

## Statistical analysis of spatial SAT scans

Kulldorff's spatial scan statistic was employed to analyze the spatial distribution of inadequate Minimum Dietary Diversity (MDD) prevalence using SAT Scan (version 9.6) and QGIS (version 3.16). Specifically, a purely spatial scan statistic was

 

utilized to detect areas where inadequate MDD was higher or lower than expected. These significant areas were visually represented using circular windows [39–42]. In the final step, we examined Poisson scan statistics. These statistics allowed us to estimate the observed and expected prevalence, as well as the Relative Risk (RR) of inadequate Minimum Dietary Diversity (MDD) within each cluster (represented by circular windows) [34].

## Multilevel mixed-effect analysis

A multi-level logistic regression analysis was performed on multilevel determinants of inadequate MDD. Due to the hierarchical nature of the DHS data, and since ICC is greater than 10% (ICC=44.1%).

ICC was calculated as follows.

$$ICC = \frac{\delta 2}{\delta 2 + \pi 2/3},$$

where δ2 indicates the estimated variance of clusters [43–45].

The log of the probability of inadequate MDD was modeled using a two-level multilevel model as follows.

$Log[\frac{\pi ij}{1-\pi ij}] = \beta 0 + \beta 1 Xij + B2\ Zij + \mu j + eij$, Where i and j are Household and community level [46] unites respectively. X and Z refer to household and community level variables respectively; πij is the probability of inadequate MDD for the ith women in the jth community; β's indicates the fixed coefficients. (B0) is the intercept, the effect on the probability of inadequate MDD in the absence of influencing factors; and μj showed the random effect (the effect of the community on the inadequate MDD of the jth community) and eij showed random errors at the household level. By assuming each community had a different intercept (B0) and fixed coefficient (β), the clustered data nature, intra, and inter-community variations were taken into account. Data is organized into, analyses of inadequate MDD (yes/no) which has the binary response and runs a mixed effect multi-level logistic regression model by considering two levels (household and community levels). During the analysis, we initially fitted a bi-variable multilevel logistic regression model and selected variables with a p-value of less than 0.2 from both Model I and Model II to develop the final model. The analysis was conducted across four distinct models. Model 0 served as the empty or null model, while Model I focused exclusively on individual-level variables. Model II examined only community-level variables, and finally, Model III integrated both community-level and individual-level variables based on a specified cutoff point. In Model III, variables with p-values less than 0.05 were deemed significantly associated with inadequate Minimum Dietary Diversity (MDD).

To assess random effects and variations, we utilized the intra-class correlation coefficient (ICC) in Model 0, the Median Odds Ratio (MOR) in Models I and II, and the Proportional Change in Variance (PCV) to illustrate variation between clusters. The ICC indicated the variation in cesarean section rates among women attributable to community characteristics, while the MOR represented the median odds ratio between areas of highest and lowest risk when randomly selecting two areas. The calculation for MOR was performed as follows:

$$MOR = exp\left(\sqrt{2 + \delta 2 + 0.6745}\right) \approx exp(2.23).$$

PCV measures the total variation attributed to individual-level variables and area-level variables in the final model (model-III). It is calculated as follows.

$$PCV = \frac{\delta 2\ of\ null\ model - \delta 2\ of\ each\ model}{\delta 2\ of\ null\ model}$$

where δ2 of the variance of the model and the variance of the null model is used as a reference.

We assessed multicollinearity among the explanatory variables by examining the standard errors at a threshold of ±2. No significant multicollinearity was detected, as the standard errors fell within this range. Additionally, we evaluated the fitness of the mixed model using the Log Likelihood Ratio or AIC/BIC criteria (as shown in Table 1).

### Ethical approval

Ethical approval was not required as we utilized data from the demographic and health survey, which anonymizes all information prior to public release, and the DHS datasets are publicly accessible. We requested an authorization letter to download the DHS dataset, which we obtained from the Central Statistical Agency (CSA) after submitting a request at https://dhsprogram.com/. The dataset and all methodologies employed in this study adhered to the guidelines established in the Declaration of Helsinki and followed DHS research protocols.

## Result

### The socio-demographic character and health service utilization of respondents

Out of 58,818 children aged 6–23, we included 57,912 weighted samples in our analysis. Among these, 37,433 children (64.64%) fell within the age range of 6–11. Additionally, 11,145 (19.24%) of their mothers were older, and 38.63% of their mothers were unable to read and write. Among the study participants, 68.64% of children had mothers residing in rural areas. Regarding maternal health service utilization, 14,703 (27%) women had both antenatal care (ANC) and postnatal care (PNC) visits (as shown in Table 2).

### Prevalence of inadequate MDD in SSA

The prevalence of MDD was 80.3% in Sub-Saharan Africa (SSA). Countries like Burkina Faso (95.5%), Ivory Coast (92.4%), Liberia (91.2%), and Chad (89.8%) had the highest prevalence of inadequate MDD. However, South Africa and Kenya demonstrated better child feeding practices, with 34.5% and 31.2% of their children receiving adequate MDD, respectively,

Across all countries, more than half of the children consumed grains, roots, and tubers, as well as breast milk (except in Gabon and South Africa). However, there were low feeding habits for legumes and nuts, eggs, and other fruits and vegetables compared to other food groups. Specifically, in terms of the consumption of different food groups: 97% of Burundi children did not consume eggs, 96% of Rwanda children did not consume eggs, 95% of Burkina Faso and Niger children did not consume eggs, 96% of Ivory Coast children did not take legumes and nuts, 97% of Guiana children did not take legumes and nuts (Table 3).

### National Gross Domestic Product (GDP) and MDD in Sub-Saharan Africa

The relationships between national levels of GDP and MDD consumption among children across countries in SSA showed that as GDP per capita increases, the prevalence of MDD consumption among children also increases. This suggests that countries with higher GDP per capita are more likely to have higher rates of MDD consumption among children (Fig 1).

**Table 1. Model fitness and selection.**

| Model | ICC | Variance | PCV | MOR | LLR | AIC | BIC |
|---|---|---|---|---|---|---|---|
| Model 0 | 22.3% | 0.9 | Reference | 6.06 | -28713 | 57430 | 57448 |
| Model 1 | | 0.83 | 27.39% | 5.65 | -366.9 | 785 | 908 |
| Model 2 | 19.76 | 1.7 | 26.09% | 5.68 | -366.6 | 783 | 902 |

**Table 2. Socio-demographic character and health service utilization of respondents in SSA from 2010–2020.**

| Variable | Category of variable | Frequency. | Percent |
|---|---|---|---|
| Maternal age | Older | 11,145 | 19.24 |
| | Middle | 20,084 | 46.08 |
| | Younger | 20,084 | 34.68 |
| children age | 12–23 | 37433 | 64.64 |
| | 6–11 | 20479 | 35.36 |
| Residence | Urban | 18,160 | 31.36 |
| | Rural | 39,752 | 68.64 |
| Educational status | No education | 22371 | 38.63 |
| | Primary | 19595 | 33.84 |
| | Secondary | 14291 | 24.68 |
| | Higher | 1650 | 2.85 |
| Media exposure | Media exposure | 36243 | 64.32 |
| | No media exposure | 20108 | 35.68 |
| ANC/PNC visit | Both | 14703 | 27.78 |
| | Either | 27933 | 52.78 |
| | Neither | 10291 | 19.44 |
| Wealth index | Poorest | 14468 | 24.98 |
| | Poorer | 12558 | 21.68 |
| | Middle | 11487 | 19.84 |
| | Richer | 10316 | 17.81 |
| | Richest | 9083 | 15.68 |

**Spatial distribution of inadequate MDD by region.** In nearly every country, the spatial distribution of inadequate minimum dietary diversity (MDD) within each country is notably high. Specifically, the following regions had a significant prevalence of inadequate MDD: Angola's Cunene region, Burkina Faso's almost all regions, Chad's Mayo Kebbi East region, Congo's Likouala region, Ethiopia's Somalia region, Guinea's Mamu region, Ivory Coast's Sud Ouest region, Lesotho's Qachis Nek region. However, some regions demonstrated relatively better MDD consumption distribution: The Nassau region in Mozambique, the Free State in South Africa, the Greater Accra region in Ghana, the Western Cape in South Africa, and the Mélange in Angola (Fig 2).

**Hot spot analysis.** The prevalence of inadequate minimum dietary diversity (MDD) clusters varies significantly both within and between countries. These clusters, representing regions or provinces, exhibit distinct patterns of MDD inadequacy. Notably, the high-inadequate MDD clusters are prevalent in most areas of Western, Northern, Eastern, and Central Sub-Saharan Africa (Fig 3).

**Spatial interpolation.** A spatial interpolation study revealed that Gabon, Cameroon, Ethiopia, the Democratic Republic of Congo, Chad, Mali, Burkina Faso, Ivory Coast, Liberia, and Senegal all faced a significant burden of inadequate minimum dietary diversity (MDD). While other countries did not fare as poorly as these did, their MDD coverage was still suboptimal (Fig 4).

**Spatial SAT scan of inadequate MDD cluster using the Poisson model.** Out of the 33 clusters analyzed using SAT Scan, only 4 exhibited statistically significant circular patterns. Cluster 1: This primary circular window was identified in Congo, Chad, Cameroon, and Gabon. Within this cluster, 4323 cases were discovered, surpassing the expected count of 3799, 93% of the children in this circular area experienced inadequate minimum dietary diversity (MDD), with a relative risk (RR) of 1.2. Cluster 2: The second significant cluster emerged in Liberia, Ivory Coast, Guinea, and neighboring countries. Here, the estimated number of inadequate MDD cases was 2763, but the observed count was 368 cases higher

**Table 3. Proportion of inadequate MDD among 6–23 month children and children's consumption of food from each food group in SSA from 2010–2020.**

| Country | Sample | Inadequate MDD | | | Breast milk | Grain root tubers | Legumes & nuts | Dairy | Flesh foods | Eggs | VA-rich FVs | Other FVs |
|---|---|---|---|---|---|---|---|---|---|---|---|---|
| | | % | No | Yes | | | | | | | | |
| Angola | 2,067 | 74.2 | 533 | 1,534 | 77 | 72 | 21 | 29 | 58 | 15 | 58 | 31 |
| Burkina-Faso | 2,062 | 95.5 | 92 | 1,970 | 93 | 62 | 7 | 12 | 21 | 5 | 23 | 5 |
| Benin | 3,865 | 75.3 | 956 | 2,909 | 80 | 54 | 25 | 31 | 48 | 22 | 31 | 27 |
| Burundi | 1,901 | 82.2 | 339 | 1,562 | 92 | 71 | 56 | 7 | 22 | 3 | 84 | 8 |
| DR. Congo | 2,567 | 84.5 | 398 | 2,169 | 88 | 65 | 12 | 9 | 49 | 9 | 65 | 26 |
| Congo | 1,488 | 84.9 | 225 | 1,263 | 62 | 85 | 10 | 48 | 60 | 8 | 48 | 12 |
| Ivory Coast | 1,110 | 92.4 | 84 | 1,026 | 75 | 83 | 4 | 15 | 55 | 10 | 17 | 11 |
| Cameroon | 1,366 | 81.3 | 256 | 1,110 | 60 | 86 | 14 | 23 | 52 | 15 | 50 | 37 |
| Ethiopia | 1,458 | 88.3 | 171 | 1,287 | 85 | 71 | 25 | 35 | 9 | 18 | 27 | 11 |
| Gabon | 1,159 | 86.5 | 156 | 1,003 | 43 | 82 | 7 | 73 | 51 | 19 | 40 | 18 |
| Ghana | 847 | 78.9 | 178 | 669 | 84 | 88 | 13 | 23 | 54 | 20 | 41 | 22 |
| Gambia | 1,167 | 81.6 | 215 | 952 | 84 | 88 | 15 | 36 | 45 | 13 | 22 | 22 |
| Guinea | 1,036 | 84.7 | 158 | 878 | 84 | 70 | 3 | 25 | 26 | 21 | 38 | 12 |
| Kenya | 2,793 | 69.8 | 843 | 1,950 | 83 | 87 | 27 | 58 | 23 | 18 | 67 | 35 |
| Comoros | 858 | 76.8 | 199 | 659 | 74 | 81 | 10 | 37 | 53 | 21 | 43 | 17 |
| Liberia | 848 | 91.2 | 74 | 774 | 80 | 74 | 7 | 16 | 45 | 9 | 39 | 12 |
| Lesotho | 464 | 86.2 | 64 | 400 | 66 | 87 | 19 | 34 | 23 | 26 | 37 | 20 |
| Mali | 2,699 | 78.7 | 573 | 2,126 | 85 | 70 | 12 | 29 | 45 | 15 | 42 | 13 |
| Malawi | 1,633 | 77.7 | 371 | 1,262 | 86 | 70 | 28 | 11 | 31 | 12 | 73 | 29 |
| Mozambique | 3,250 | 75.4 | 799 | 2,451 | 83 | 79 | 28 | 13 | 42 | 18 | 60 | 33 |
| Nigeria | 3,575 | 76.2 | 852 | 2,723 | 70 | 84 | 35 | 31 | 39 | 18 | 43 | 16 |
| Niger | 1,554 | 88.7 | 176 | 1,378 | 86 | 75 | 13 | 18 | 15 | 5 | 28 | 7 |
| Namibia | 649 | 77.7 | 145 | 504 | 58 | 65 | 10 | 35 | 62 | 22 | 38 | 30 |
| Rwanda | 1,138 | 71.7 | 329 | 809 | 93 | 71 | 66 | 22 | 18 | 4 | 70 | 26 |
| Sierra Leone | 1,440 | 74.5 | 373 | 1,067 | 76 | 79 | 18 | 33 | 47 | 17 | 45 | 22 |
| Senegal | 1,757 | 82.2 | 314 | 1,443 | 82 | 82 | 9 | 38 | 44 | 9 | 45 | 8 |
| Chad | 2,927 | 89.8 | 299 | 2,628 | 85 | 61 | 8 | 26 | 34 | 8 | 25 | 10 |
| Togo | 1,062 | 79.2 | 221 | 841 | 87 | 82 | 17 | 11 | 56 | 11 | 54 | 10 |
| Tanzania | 3,014 | 78.8 | 640 | 2,374 | 81 | 91 | 37 | 23 | 32 | 7 | 66 | 20 |
| Uganda | 1,366 | 76.8 | 317 | 1,049 | 77 | 81 | 52 | 30 | 33 | 14 | 50 | 21 |
| South Africa | 428 | 64.5 | 152 | 276 | 40 | 86 | 11 | 74 | 45 | 41 | 45 | 40 |
| Zambia | 2,771 | 79.4 | 571 | 2,200 | 74 | 86 | 23 | 12 | 41 | 23 | 64 | 27 |
| Zimbabwe | 1,593 | 77.3 | 362 | 1,231 | 69 | 94 | 20 | 21 | 43 | 16 | 60 | 27 |
| Total(SSA) | 57, 912 | 80.3 | 11,435 | 46,477 | 80 | 76 | 22 | 26 | 38 | 14 | 48 | 20 |

than expected. The relative risk was 1.1, and 92% of the children within this circular window faced inadequate MDD. Cluster 3: Another noteworthy circular window was detected in Burkina Faso. Although the predicted number of cases in this cluster was 1164, the actual count was 1385. The relative risk stood at 1.2, and 97% of the children in this area were at risk of inadequate MDD. Cluster 4: The fourth most significant circular window was found in Ethiopia. Within this cluster, 856 cases were observed, exceeding the expected count of 736. The relative risk was 1.2, and 95% of the children in this region faced the risk of inadequate MDD (Fig 5).

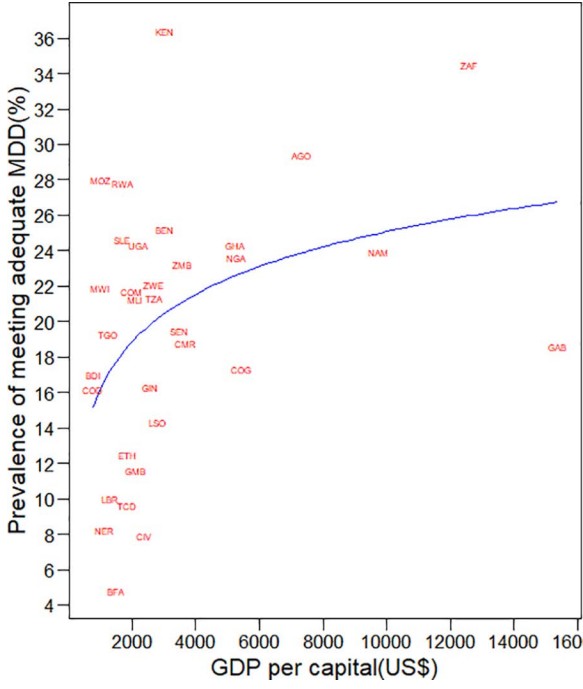

**Fig 1. The relationship of MDD consumption among children 6–23 months and GDP per capita in countries of SSA.**

**The consequences of inadequate MDD.** The logistic regression analysis indicated that children who do not consume adequate MDD are 2.5 times more likely to experience anemia, 1.30 times more likely to experience wasting, and 1.12 times more likely to experience stunting compared to their counterparts (Fig 6).

## Multilevel mixed effect logistic regression analysis for the multilevel factors of inadequate MDD among children in SSA

After adjusting for other household-level factors such as child age, mother's age, media exposure, ANC, PNC, maternal working status, maternal and partner education level, and maternal stunting and wasting, as well as community-level factors like household wealth and distance to health facilities, several statistically significant predictors emerged.

Children aged 6–11 were twice as likely to have inadequate MDD compared to children aged 12–23 [Adjusted Odds Ratio [AOR= 2.0, 95%CI (1.8–2.2)]. Children born to mothers aged 25–34 were 20% less likely to experience inadequate MDD compared to those born to mothers aged 35–49 [AOR= 0.8, 95%CI (0.8–0.98)]. Children born to mothers with secondary and higher education were 33% and 49% less likely, respectively, to have inadequate MDD compared to those born to mothers with no education [AOR= 0.77, 95% CI (0.67–0.87)] and [AOR: 0.51, 95% CI (0.4–0.6)]. Children born to mothers who had either ANC or PNC (but not both) were 1.2 and 1.4 times more likely, respectively, to experience inadequate MDD compared to those born to mothers who had both ANC and PNC [AOR= 1.2, 95% CI (1.06–1.28)] and [AOR= 1.4, 95% CI (1.2–1.5)]. Children born to mothers with no media exposure were 1.4 times more likely to have inadequate MDD compared to those with media exposure [AOR= 1.38, 95% CI (1.19–1.52)]. Children born to families with middle, richer, and richest wealth were 18%, 39%, and 58% less likely, respectively, to experience inadequate MDD compared to those from the poorest wealth category [AOR= 0.82, 95%CI (0.7–0.94)], [AOR= 0.6, 95%CI (0.5–0.7)], and [AOR= 0.4, 95%CI (0.36–0.5)]. Children born to mothers who were stunting and wasting were 1.4 times more likely to have inadequate MDD compared to those born to mothers who were not stunting and wasting [AOR= 1.4, 95% CI (1.01–2.04)].

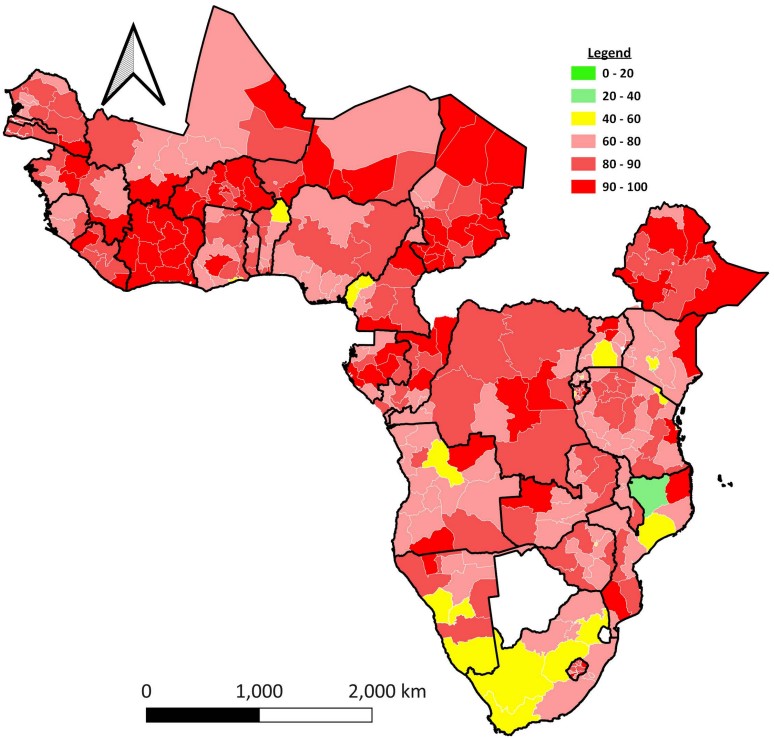

**Fig 2. The spatial distribution of inadequate MDD among children 6–23 months in each region (admin 1) of SSA countries.**

Children born to families with middle, richer, and richest wealth were 18%, 39%, and 58% less likely, respectively, to experience inadequate MDD compared to those from the poorest wealth category [AOR= 0.82, 95%CI (0.7–0.94)], [AOR= 0.6, 95%CI (0.5–0.7)], and [AOR= 0.4, 95%CI (0.36–0.5)]. Children born to mothers who were stunting and wasting were 1.4 times more likely to have inadequate MDD compared to those born to mothers who were not stunting and wasting [AOR= 1.4, 95%CI (1.01–2.04)]. Relatively small distance to health facilities was linked to lower odds of inadequate MDD [AOR= 0.8, 95%CI (0.7–0.9)] (see Table 4).

The first model was the model-0= empty model or null model was conducted without an independent variable (univariate analysis) and the result showed intra-class correlation (ICC) = 22.3%, the second model was model 1= analysing only individual-level variable, the 3rd model was model-2 (analysing only community-level variable), the last model was model 3= analysing both individual and community-level variable.

## Discussion

Analyzing regional, national, and continental level data on MDD can offer valuable insights for creating contextually relevant strategies and policies in support of SDG 2 target 2.2, which aims to eradicate all forms of malnutrition. Therefore this study analyzed geospatial distribution and multilevel factors of inadequate minimum dietary diversity and its consequences among children in SSA.

The prevalence of inadequate MDD in SSA was 80.3%. Notably, all SSA countries exhibited inadequate MDD coverage, which is a higher prevalence compared to other previous study findings in SSA [19,47] and study findings in Indonesia [48] and Brazil [49]. This situation might be attributed to insufficient child health services in the area, including complementary feeding practices [50]. Another factor to consider is that food costs can significantly influence dietary diversity [51, 52] as evidenced by the relationship between gross domestic product and MDD consumption in different countries in SSA.

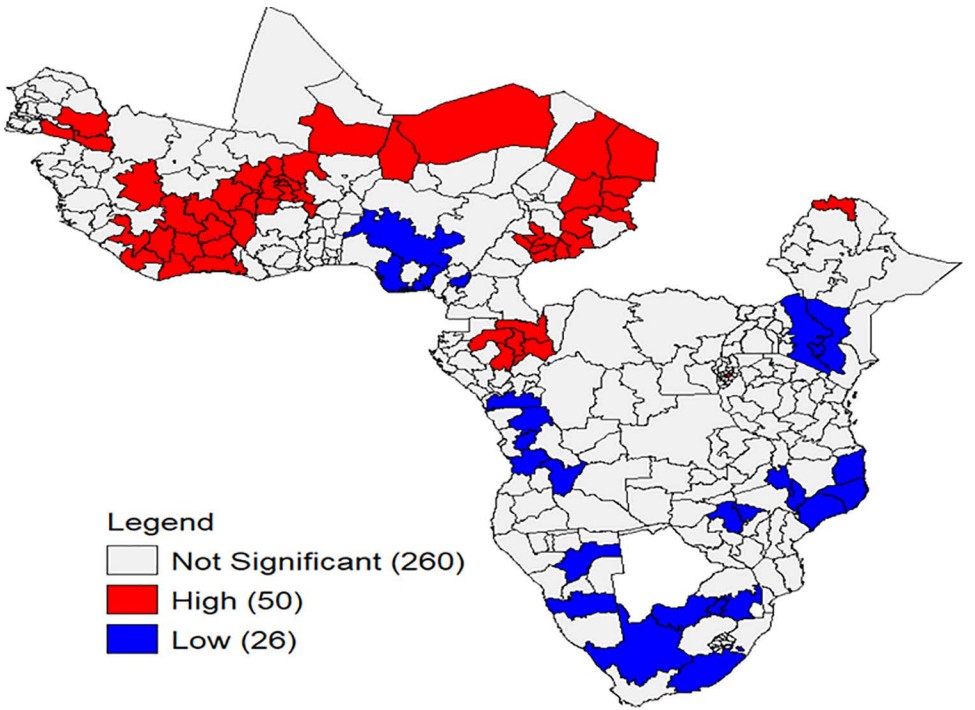

**Fig 3. The spatial hot spot analysis of inadequate Minimum Dietary Diversity (MDD) among children aged 6–23 months in Sub-Saharan Africa (SSA).** Each polygon on the map corresponds to a single administrative region or province within SSA. The color scheme indicates the level of MDD inadequacy: High (Red): represents areas with a high rate of inadequate MDD, forming hot spots. Low (Blue): indicates regions with a low rate of inadequate MDD, forming cold spots.

Children in Burkina Faso, Ivory Coast, Ethiopia, Liberia, Lesotho, Niger, and Chad had inadequate MDD. Spatial variation occurs both between and within countries. This finding supported the previous study finding in these countries Burkina Faso [53], Ivory Coast [54], Ethiopia [55], and Liberia [56]. This will be tied to various dimensions such as agriculture, market availability, equity of health care, education, and sub-national healthcare disparities [57]. Therefore, it suggests that it is important to need geographically targeted interventions to address the underlying drivers of inadequate MDD in these high-burden areas.

Cereals and tubers are the most commonly consumed foods in this region, which aligns with findings from other studies [58–60]. However, when compared to other regions, SSA has a significantly lower consumption rate [61, 62]. This indicates that nutrition in SSA is influenced by rapidly changing political, socioeconomic, drought, and insecurity landscapes, as well as other natural and artificial occurrences [63].

The consumption of eggs, fruits, and vegetables in this study was indeed low. This finding aligns with other research as well [64]. This phenomenon can be attributed to financial constraints or the fact that individuals in low-income countries tend to consume fewer fruits and vegetables [52,64]. Compared to other dietary groups, the consumption of eggs and other fruits and vegetables is notably low. One potential reason for this disparity is that individuals in rural areas are less inclined to consume eggs due to economic challenges. They often choose to sell eggs and purchase alternative, more affordable food options as a way to mitigate economic difficulties.

The spatial distribution of inadequate MDD differs widely throughout SSA countries. Based on the geographic analysis, regions in the western SSA and the northern areas of Central and East SSA are at a high risk of inadequate dietary diversity among children. This is supported by previous study findings in SSA [65]. These variations may stem from factors such as environmental shifts, geographic differences, topographical variations, socioeconomic disparities, and variations

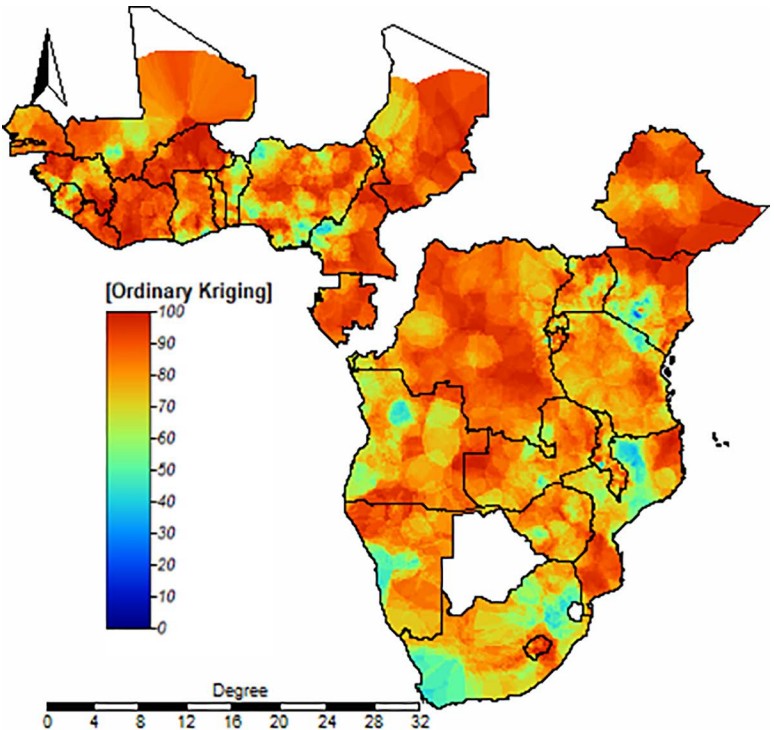

**Fig 4. The interpolation of inadequate MDD among children 6–23 months in SSA.** The interpolated continuous images were made in inadequate MDD using standard ordinary kriging interpolation. The transition from bold blue to bold red reflects an increase in the burden of inadequate MDD.

in healthcare services. These findings provide insights into regions where interventions are needed to address the nutritional challenges faced by young children in SSA.

Children under one year of age experience inadequate MDD compared to their counterparts. This is similar to other findings [66, 67]. This may be attributed to the fact that starting at 6 months, infants can consume pureed, mashed, and semi-solid foods.

Previous research aligns with our findings, indicating that both maternal and partner education exhibit an inverse relationship with inadequate maternal MDD [3,61,68,69]. One potential explanation is that educated parents, both fathers and mothers, are better equipped to comprehend educational messages disseminated through mass media channels like radio or newspapers. This understanding enables them to adopt positive practices related to ensuring MDD for their children. Additionally, maternal knowledge and willingness to make informed care decisions may also play a role [70].

Children born to mothers who received both ANC and PNC visits have a higher likelihood of achieving adequate MDD compared to children born to mothers who received either ANC or PNC or neither. This finding aligns with previous research results [58,71]. This is because mothers who receive both ANC and PNC are more likely to receive nutrition education on topics such as breastfeeding, complementary feeding, and dietary diversity.

Maternal stunting and wasting are closely linked to inadequate MDD. This finding is in line with another result [17,72]. This may be due to maternal stunting and wasting resulting from chronic malnutrition, which can lead to deficiencies in essential nutrients. When mothers lack proper nutrition, they may not be able to provide sufficient nutrients to their infants during pregnancy and breastfeeding [73]. Inadequate breastfeeding can limit the introduction of diverse foods to the child, affecting MDD [73].

According to a study, households with higher incomes had a higher consumption of MDD in contrast to poorer ones. This study highlights the correlation between income and consumption of MDD among children. This finding was

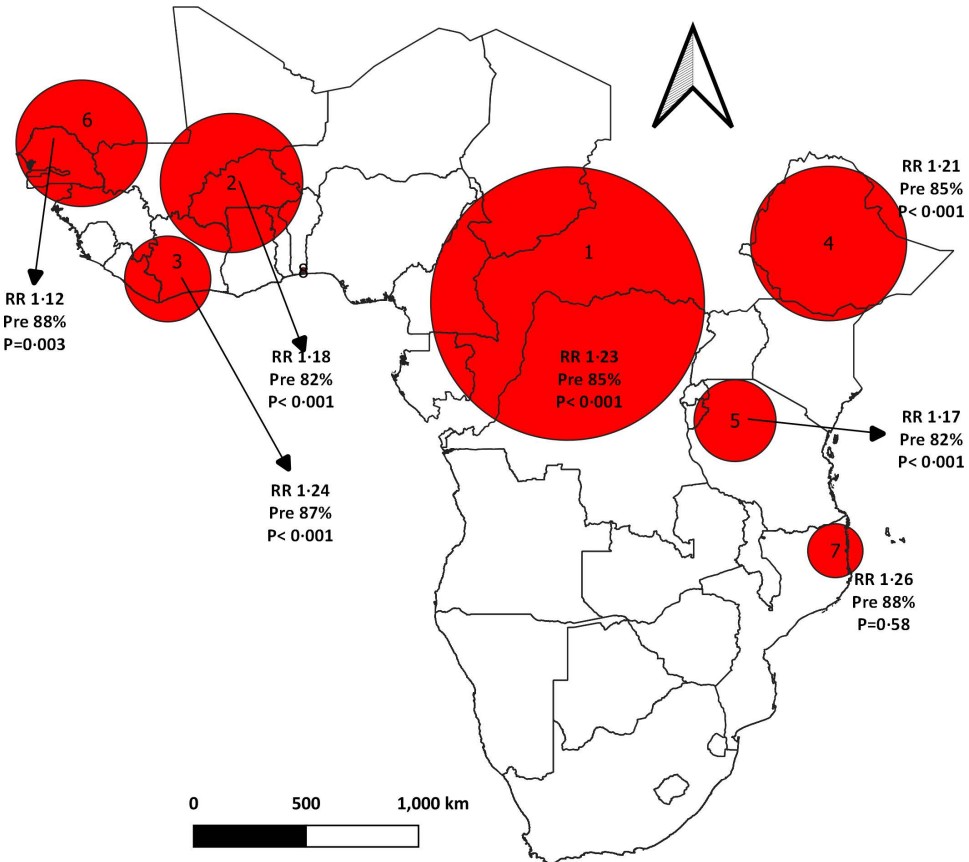

**Fig 5. The spatial scan statistics analysis of inadequate Minimum Dietary Diversity (MDD) among children aged 6–23 months in Sub-Saharan Africa (SSA) using the Poisson model.** The analysis identified red circular windows that represent hotspot areas with high rates of inadequate MDD CI: Cluster number corresponding to the circular window on the map. O: Observed cases of inadequate MDD, E: Expected cases of inadequate MDD, RR: Relative risk, Pre: Prevalence, P: p-value abbreviations were used.

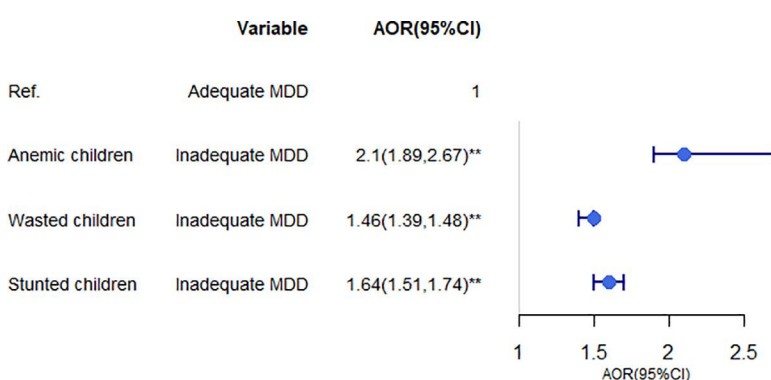

**Fig 6. Logistic regression analysis of the consequences of inadequate MDD among children aged 6–23 months in SSA.**

**Table 4. Multilevel regression analysis of multilevel determinants of inadequate MDD among children in SSA Using DHS from 2010–2020.**

| Variable | Category | Bivariable analysis | Null Model (Model 0) | Model 1 | Model 2 | Final model |
|---|---|---|---|---|---|---|
| | | COR | ICC=22.3% | | AOR | AOR |
| Child age | 12—23 | 1 | | 1 | | 1 |
| | 6–11 | 1.8(1.7–1.9) | | 2(1.9,2.2) | | 2(1.84,2.18)* |
| Mother age | 35—49 | 1 | | 1 | | 1 |
| | Middle 25–34 | 0.9(0.85–0.95) | | 0.9(0.8,0.9) | | 0.89(0.8,0.98)* |
| | Younger(<25) | 1.01(1.01–1.13) | | 1.08(1,1.2) | | 1.03(0.92,1.15) |
| Media | (Ref. Exposed) | 1 | | 1 | | 1 |
| | Not Exposed | 1.9(1.8–1.97) | | 1.9(1.8,2) | | 1.38(1.26,1.52)* |
| ANC/PNC | (Ref. Both) | 1 | | 1 | | 1 |
| | Either | 1.36(1.3–1.4) | | 1.4(1.3,1.5) | | 1.17(1.06,1.28)* |
| | Neither | 1.6(1.5–1.7) | | 1.7(1.5,1.8) | | 1.35(1.19,1.52)* |
| Mother work | Have no work | 1 | | 1 | | 1 |
| | Have work | 0.85(0.8–0.89 | | 0.7(0.7,0.8) | | 0.81(0.74,0.88)* |
| Maternal educational status | No education | 1 | | 1 | | 1 |
| | Primary | 0.75(0.72–0.79) | | 0.8(0.7,0.8) | | 0.93(0.83,1.04) |
| | Secondary | 0.52(0.49–0.55) | | 0.5(0.5,0.6) | | 0.77(0.67,0.87)* |
| | Higher | 0.22(0.2–0.24) | | 0.2(0.2,0.3) | | 0.51(0.4,0.64)* |
| Sex of household head | Male | 1 | | 1 | | 1 |
| | Female | 1.02(0.97–1.07) | | 1(1.02,1.16) | | 1.05(0.94,1.17) |
| Birth in last 5 years | One | 1 | | 1 | | 1 |
| | Two | 1.2(1.15–1.25) | | 1.2(1.1,1.2) | | 0.99(0.91,1.08) |
| | above two | 1.47(1.34–1.62) | | 1.3(1.2,1.5) | | 0.99(0.84,1.18) |
| Partner Education | No education | | | 1 | | 1 |
| | Primary | 0.75(0.72–0.79) | | 0.65(0.6,0.7) | | 1.02(0.9,1.14) |
| | Secondary | 0.52(0.49–0.55) | | 0.5(0.46,0.54) | | 0.99 (0.87,1.12) |
| | Higher | 0.22(0.2–0.24) | | 0.26(0.23,0.29) | | 0.82(0.68,0.98)* |
| Mother stunting and wasting. | Normal | 1 | | 1 | | 1 |
| | Stunted | 1.03(0.96–1.06) | | 1.1(1,1.2) | | 1.09(0.98,1.23) |
| | Wasted | 1.43(1.28–1.6) | | 1.4(1.2,1.6) | | 1.11(0.93,1.32) |
| | Overlap | 1.78(1.37–2.3) | | 1.7(1.3,2.2) | | 1.43(1.01,2.04)* |
| Wealth | Poorest | 1 | | 1 | | 1 |
| | Poorer | 0.8(0.77–0.88) | | 0.8(0.8,0.9) | | 0.92(0.81,1.05) |
| | Middle | 0.65(0.6–0.69) | | 0.7(0.6,0.7) | | 0.82(0.72,0.94)* |
| | Richer | 0.5(0.47–0.54) | | 0.5(0.4,0.5) | | 0.61(0.53,0.7)* |
| | Richest | 0.33(0.3–0.35) | | 0.3(0.3,0.31) | | 0.42(0.36,0.5)* |
| Residence | Urban | 1 | | | 1 | 1 |
| | Rural | 1.8(1.7–1.9) | | | 2.3(2.1,2.5) | 1.05(0.92,1.19) |
| Ecology | Highland/>2300 | 1 | | | 1 | 1 |
| | Temperate(1501–2300 masl) | 0.59(0.44–0.79) | | | 0.7(0.5,1) | 0.91(0.62,1.33) |
| | Lowland(501–1500 masl) | 0.64(0.48–0.83) | | | 0.7(0.5,0.9) | 0.83(0.57,1.2) |
| | Subtropical(<501 masl) | 0.62(0.46–0.83) | | | 0.7(0.5,0.9) | 0.84(0.58,1.21) |
| Distance of health facility | Big problem | | | | 1 | 1 |
| | Not big problem | 0.7(0.69–0.75) | | | 0.7(0.7,0.8) | 0.84(0.77,0.92)* |

Key: * statistically significant at p - values <0·05; ANC=Antenatal Care; PNC=Postnatal Care 1= reference.

supported by other study findings in Ethiopia [74], Ghana [75], and all SSA countries [76]. The possible reason may be children from poor households are less likely to eat fruits and vegetables because of economic problems and they may sell fruits and vegetables and buy other food kinds with lower cost to lessen the economic problem [77].

Media exposure has been associated with a reduced risk of inadequacy in MDD. This relationship may be attributed to mothers who have greater access to information about MDD. Furthermore, this connection could be linked to the generation and dissemination of knowledge regarding dietary diversity and promoting healthcare practice.

Longer distance to health facilities was associated with increases in the risk of inadequate MDD among children. This is supported by the study conducted in Malawi [78]. This may be because, when health facilities are located far from communities, it becomes challenging for caregivers, especially mothers, to access them. Longer distances may discourage regular visits for health check-ups, counseling, and nutrition education. As a result, caregivers may lack awareness of optimal feeding practices, leading to inadequate MDD for their children [79]. Addressing geographical barriers, improving transportation options, and enhancing health facility quality are essential to promote optimal child feeding practices, even in areas with longer distances to health facilities.

This study indicates that in children who did not consume adequate MDD, anemia, stunting, and wasting are significantly associated. This finding is supported by other study findings [80–82]. This may be because children, who do not consume adequate MDD, it suggests that they are not receiving a sufficient variety of nutrients essential for growth and development. Inadequate MDD can contribute to iron deficiency anemia, wasting, and stunting in children due to insufficient intake of essential nutrients needed for proper growth and maintenance of body tissues [83].

### Practical and policy implications

The high prevalence of inadequate MDD in SSA highlights the need for targeted nutrition policies and programs to improve dietary practices among children aged 6–23 months.

The spatial clustering of inadequate MDD in certain regions of SSA, such as Gabon, Cameroon, Ethiopia, Chad, Mali, Burkina Faso, Ivory Coast, Liberia, and Senegal, suggests the need for geographically targeted interventions to address the underlying drivers of inadequate MDD in these high-burden areas.

The identified multilevel factors associated with inadequate MDD, including maternal age, education, antenatal/postnatal care, media exposure, wealth status, maternal nutritional status, and distance to health facilities, should be considered in the design and implementation of comprehensive nutrition policies and programs.

### Strength and limitation

The study employed geospatial analysis to examine the spatial distribution of inadequate MDD among children in SSA and the study utilized multilevel analysis to examine the determinants of inadequate MDD could be the strength of the study. The enumeration areas are not visible on the interpolated map, because the point of the enumeration area of some countries was dense, all areas under prediction are hidden and the DHS data was collected in a variety of years, depending on the country could be the limited of the study.

### Conclusion

In Sub-Saharan Africa, the prevalence of inadequate MDD is notably high, particularly in Burkina Faso and Ivory Coast. Geospatial distribution highlights that countries such as Gabon, Cameroon, Ethiopia, the Democratic Republic of Congo, Chad, Mali, Burkina Faso, Ivory Coast, Liberia, and Senegal face a significant burden of inadequate MDD. Maternal age, educational status, occupational status, and nutritional status of the mother (such as stunting and wasting), children's age, antenatal and postnatal care visits, along with community-level factors like media exposure, wealth index, and proximity to health facilities, are associated with inadequate MDD. The spatial clustering of inadequate MDD in certain regions of SSA, suggests the need for geographically targeted interventions to address the determinants of inadequate MDD in

these high-burden areas. Therefore, strategies aimed at promoting frequent antenatal and postnatal visits and improving maternal nutrition are crucial for reducing inadequate MDD among children. This study highlights a significant association between MDD and anemia, stunting, and wasting in children aged 6–23 months. To address these critical issues, it is essential to improve MDD among children, as this intervention can play a vital role in achieving SDG target 2.2, which aims to end all forms of malnutrition by 2030.

## Supporting information

**S1 Table. The country included in the analysis using DHS data from 2010–2020.**
(DOCX)

## Acknowledgments

The authors are sincerely grateful to the Demographic Health Survey (DHS) program for allowing us to use the DHS dataset through their archives (https://dhsprogram.com).

## Author contributions

**Conceptualization:** Bayuh Asmamaw Hailu, Chala Daba, Yawkal Tsega, Abel Endawkie.

**Data curation:** Bayuh Asmamaw Hailu, Abel Endawkie.

**Formal analysis:** Bayuh Asmamaw Hailu, Abel Endawkie.

**Investigation:** Bayuh Asmamaw Hailu, Abel Endawkie.

**Methodology:** Bayuh Asmamaw Hailu, Chala Daba, Yawkal Tsega, Abel Endawkie.

**Resources:** Bayuh Asmamaw Hailu.

**Software:** Bayuh Asmamaw Hailu, Abel Endawkie.

**Supervision:** Bayuh Asmamaw Hailu, Abel Endawkie.

**Validation:** Bayuh Asmamaw Hailu, Chala Daba, Yawkal Tsega, Abel Endawkie.

**Visualization:** Bayuh Asmamaw Hailu, Chala Daba, Yawkal Tsega, Abel Endawkie.

**Writing – original draft:** Bayuh Asmamaw Hailu, Chala Daba, Yawkal Tsega, Ashebir Asaminew, Birhan Asmame Miheretu, Abel Endawkie.

**Writing – review & editing:** Bayuh Asmamaw Hailu, Chala Daba, Yawkal Tsega, Ashebir Asaminew, Birhan Asmame Miheretu, Abel Endawkie.

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
