## [Decision Letter · Decision Letter 0]

21 Jun 2024

PONE-D-24-12330Spatial Distribution, Multilevel Factorsof Inadequate Minimum Dietary Diversity, and its Consequence among Children aged 6-23 months in Sub-Saharan Africa:PLOS ONE

Dear Dr. Endawkie,

Thank you for submitting your manuscript to PLOS ONE. After careful consideration, we feel that it has merit but does not fully meet PLOS ONE’s publication criteria as it currently stands. Therefore, we invite you to submit a revised version of the manuscript that addresses the points raised during the review process.

We look forward to receiving your revised manuscript.

Kind regards,

Jay Saha

Academic Editor

PLOS ONE

Reviewers' comments:

Reviewer's Responses to Questions

**Comments to the Author**

1. Is the manuscript technically sound, and do the data support the conclusions?

Reviewer #1: Yes

2. Has the statistical analysis been performed appropriately and rigorously? 

Reviewer #1: Yes

3. Have the authors made all data underlying the findings in their manuscript fully available?

Reviewer #1: Yes

4. Is the manuscript presented in an intelligible fashion and written in standard English?

Reviewer #1: No

5. Review Comments to the Author

Reviewer #1: General comments

Although writing an English scientific manuscript is a challenge for many scholars, especially for non-native speakers, I appreciate the authors' efforts. However, there are many grammatical errors and language across the document. As a result in the scholarly community, it is important to maintain the message being conveyed in the manuscript is unambiguous as possible. Hence, your manuscript must be reviewed by a person proficient in written English.

The research topic, "Spatial Distribution, Multilevel Factors of Inadequate Minimum Dietary Diversity, and its Consequence among Children aged 6-23 months in Sub-Saharan Africa," aligns well with the findings of the manuscript. The study effectively addresses the spatial distribution and multilevel factors, and it also you tried to identify significant health consequences of inadequate dietary diversity. However, the consequences are not as elaborately discussed as the other aspects, which might lead to a perception that the focus was more on factors and distribution. If the consequences were intended to be a primary focus, they could be emphasized more in the discussion and conclusion sections to better align with the stated topic. Otherwise I suggest revision of the topic.

Abstract

Abstract, Line 15-16: The phrase "Because of the evidence of geospatial distribution and multilevel factors of inadequate minimum dietary diversity (MDD) and its consequence among children in Sub-Saharan Africa (SSA) remains limited and inadequate feeding practices can lead to malnutrition in young children" need to be rephrased for clarity.

Introduction

The introduction is comprehensive but somewhat dense. While it covers the necessary background, the flow can be improved to enhance readability. Breaking down long sentences

For instance, the first sentence could be split into two: "Adequate minimum dietary diversity (MDD) is crucial for the healthy growth and development of children aged 6 to 23 months. Nutrient adequacy of a diet is influenced by food groupings."

The introduction does a good job of explaining why MDD is important and the consequences of inadequate dietary diversity. However, it would benefit from more specific context about why this study is necessary. What gaps in the existing literature does this study address?

The connection to the SDGs is a strong point but needs to be more explicitly tied to the study's objectives. While the introduction mentions SDG 2, it could be more explicit about how the findings of this study will contribute to achieving these goals.

The introduction uses various statistics to highlight the prevalence of malnutrition and inadequate MDD. Ensure that all statistics are up-to-date and properly referenced.

Ensure all terms and abbreviations are clearly defined upon first use. For instance, while MDD is defined, terms like ANC (antenatal care) and PNC (postnatal care) are not immediately explained. A brief definition of these terms when first mentioned would aid in clarity.

The introduction should aim to engage the reader by highlighting the potential impact of the study. Emphasizing how this research can lead to actionable strategies and policy recommendations will underline its importance.

Methods

Methods, Line 21: Specify the tools or software used for spatial analysis

The manuscript outlines that data were sourced from multiple cross-sectional surveys conducted in SSA in different years. While the nature of surveys are mentioned, there is a lack of detail on how the data from these surveys were harmonized to ensure consistency and comparability. Range of years, information on the survey instruments, sampling methods, and the measures taken to address potential biases in the data collection process would be beneficial. Additionally, details on the specific demographic and geographic variables collected should be provided to understand the data's scope fully.

The study employs a spatial scan statistic to analyze the geographical distribution of your outcome variable in SSA. The chosen methodology allows for the identification of spatial clusters, which is appropriate for the study's aim of detecting geographical patterns. However, additional clarity on the rationale behind selecting this specific statistical method over others would enhance the robustness of the justification. For instance, a comparison with other clustering methods like Bayesian approaches could provide a more comprehensive background for this study.

Discussion

The discussion rightly connects the high prevalence of inadequate MDD to economic factors and healthcare service disparities. However, it would be beneficial to provide more detailed comparisons with specific regions outside SSA to highlight unique challenges or similarities. This can help in understanding whether similar interventions might be applicable or if entirely new strategies are needed.

The spatial analysis is a strong point of the discussion, as it highlights the importance of localized interventions. It would be useful to include more specific examples or case studies from the mentioned countries (e.g., Burkina Faso, Ivory Coast) to illustrate how these problem manifest differently and what specific strategies have been or could be employed effectively.

Line 300-306, you stated that, the low consumption of eggs, fruits, and vegetables is attributed to financial constraints and economic challenges in rural areas, where people might prefer selling nutritious food items like eggs to mitigate economic difficulties. This section should be effectively points out the economic rationale behind dietary choices. To strengthen this point, the discussion could benefit from integrating qualitative data or anecdotes from similar studies that capture the lived experiences of families making these choices. This would provide a more holistic understanding of the barriers to adequate nutrition.

The policy recommendations are crucial, and the call for improved public education is particularly relevant. However, the discussion could be enriched by suggesting specific types of educational programs or policy interventions that have proven successful in similar contexts.

6. PLOS authors have the option to publish the peer review history of their article (what does this mean? ). If published, this will include your full peer review and any attached files.

**Do you want your identity to be public for this peer review?** For information about this choice, including consent withdrawal, please see our Privacy Policy .

Reviewer #1: No

---

## [Author Response · Author response to Decision Letter 1]

25 Jun 2024

Authors’ Responses to Editor and Reviewers Comments

Dear Academic Editor of PLOS One

“Geospatial Distribution, Multilevel Factors of Inadequate Minimum Dietary Diversity, and its Consequence among Children aged 6-23 months in Sub-Saharan Africa”

Dear PLOS One Editors and Reviewers;

We are thankful for your constructive comments. We have looked at the comments and have revised our paper accordingly. We hope our paper improved as a result of incorporating the reviewer’s and academic editor’s comments and suggestions. Here are the authors’ responses to the comments.

Please find for your kind consideration the following:

1. A revised manuscript without track changes.

2. A revised paper with tracked changes

3. A rebuttal letter that responds to each point raised by the academic editor and reviewer.

The point-by-point responses of authors are written by hoping these changes would meet with your favorable consideration, we are happy to hear if there are more comments and suggestions. Please do not hesitate to let us know if you have any questions.

Yours Sincerely

Mr. Abel Endawkie Correspondence Author

Department of Epidemiology and Biostatistics School of Public Health College of Medicine and Health Science Wollo University Dessie Ethiopia

Tel. 251935459310 Email address abelendawkie@gmail.com

We have tried our best to improve it accordingly:

Please revise the manuscript.

Point-by-point response of Authors for editor's and reviewers' comment

Editor's comments to the Authors

https://journals.plos.org/plosone/s/file?id=wjVg/PLOSOne_formatting_sample_main_body.pdf andhttps://journals.plos.org/plosone/s/file?id=ba62/PLOSOne_formatting_sample_title_authors_affiliations.pdf

Response: We thank you and appreciate your prompt and timely response. To ensure compliance with PLOS ONE's formatting guidelines, we have meticulously followed the typesetting requirements for references, tables, and technical specifications for figures and to expedite the process and prevent any delays.

Reviewer comment for Authors

General comments

Although writing an English scientific manuscript is a challenge for many scholars, especially for non-native speakers, I appreciate the authors' efforts. However, there are many grammatical errors and language across the document. As a result in the scholarly community, it is important to maintain the message being conveyed in the manuscript is unambiguous as possible. Hence, your manuscript must be reviewed by a person proficient in written English.

Response: Dear reviewer; we are great full to thank for your kind, prompt, and timely response. To the maximum effort of authors and grammar and spelling checkup, we tried to correct your suggestion of English writing of scientific manuscript.

Comment 1: The research topic: "Spatial Distribution, Multilevel Factors of Inadequate Minimum Dietary Diversity, and its Consequence among Children aged 6-23 months in Sub-Saharan Africa," aligns well with the findings of the manuscript. The study effectively addresses the spatial distribution and multilevel factors, and it also you tried to identify significant health consequences of inadequate dietary diversity. However, the consequences are not as elaborately discussed as the other aspects, which might lead to a perception that the focus was more on factors and distribution. If the consequences were intended to be a primary focus, they could be emphasized more in the discussion and conclusion sections to better align with the stated topic. Otherwise I suggest revision of the topic.

Response 1: We sincerely appreciate your insightful comment and suggestion and we apologize for any confusion in our writing. Our study focuses on the Spatial Distribution, Multilevel Factors of Inadequate Minimum Dietary Diversity, and for policy implication we want to highlight the consequence of inadequate minimum dietary diversity among children 6-23 months age. We indicated that in discussion part: Children who did not consume adequate MDD, anemia, stunting, and wasting are significantly associated. This is finding is supported by other study findings (73-75). This may be because children, who do not consume adequate MDD, it suggests that they are not receiving a sufficient variety of nutrients essential for growth and development. Inadequate MDD can contribute to iron deficiency anemia, wasting and stunting in children due to insufficient intake of essential nutrients needed for proper growth and maintenance of body tissues (76).

Conclusion: This study indicates that in children who did not consume adequate MDD, anemia, stunting, and wasting are significantly associated. Therefore, improving adequate MDD working at determinants of MDD among children helps to achieve the SDG target 2.2 to end all forms of malnutrition by 2030.

Abstract

Comment 2: Abstract, Line 15-16: The phrase "Because of the evidence of geospatial distribution and multilevel factors of inadequate minimum dietary diversity (MDD) and its consequence among children in Sub-Saharan Africa (SSA) remains limited and inadequate feeding practices can lead to malnutrition in young children" need to be rephrased for clarity.

Response 2: We would like to thank you for your deep comment. We appreciated your efforts in reviewing the content. We would like to apologize for we confused you in writing. We had revised and corrected it.

Introduction

Comment 3: The introduction is comprehensive but somewhat dense. While it covers the necessary background, the flow can be improved to enhance readability. Breaking down long sentences

Response 3: We would like to thank you for your deep comment. We appreciated your efforts in reviewing the content. We would like to apologize for we confused you in writing. We had revised and corrected it.

Comment 4: For instance, the first sentence could be split into two: "Adequate minimum dietary diversity (MDD) is crucial for the healthy growth and development of children aged 6 to 23 months. Nutrient adequacy of a diet is influenced by food groupings."

Response 4: We would like to thank you for your deep comment. We appreciated your efforts in reviewing the content. We would like to apologize for we confused you in writing. We had revised and corrected it.

Comment 5: The introduction does a good job of explaining why MDD is important and the consequences of inadequate dietary diversity. However, it would benefit from more specific context about why this study is necessary. What gaps in the existing literature does this study address?

Response 5: We would like to thank you for your deep comment. We appreciated your efforts in reviewing the content. We would like to apologize for we confused you in writing. We had revised and corrected it. Here are the existing evidence and the reason for conducted this research.

Existed evidence: SDG 2 focuses on achieving food security and improving nutrition through universal access to safe and nutritious food to achieve SDG target 2.2 to end all forms of malnutrition (14). However, in SSA, the situation remains contrary to these goals, with statistics showing a rise in malnutrition (15).

Despite the recommendation and significant advances in child nutrition, the prevalence of inadequate MDD is significantly high in SSA and ranged from 67% to 94% in SSA.

Different literatures showed that, the women's education (19-21), the husband's educational status (19-22), child age (20, 23), mass media exposure (19-21), maternal age (23, 24), occupational status (25, 26), wealth index (19-21), antenatal care (22, 27), and residence (26) in were found to be associated with MDD among children aged 6 to 23 months in different countries of SSA.

Importance of study: The evidence of geospatial distribution both within and between countries, along with the multilevel factors of inadequate MDD and its consequence among children aged 6 - 23 months are important to meet the expected SDG 2.2 by 2030 (end all types of malnutrition). It provides space specific evidence, which enables national and sub-national comparisons, tracks changes over time, and identify the high-risk populations. It informs targeted interventions, guides policy decisions, and monitors progress toward nutrition objectives set by the WHO and SDG in Africa.

Gap identified: However, as far as the author searching effort, the evidence of geospatial distribution between/within countries, and multilevel factors of inadequate MDD and its consequence among children aged 6-23 months in SSA are remains limited. Therefore, this study aimed to identify geospatial distribution and multilevel factors of inadequate MDD and its consequence among children aged 6 to 23 months in SSA using recent DHS data.

Comment 6: The connection to the SDGs is a strong point but needs to be more explicitly tied to the study's objectives. While the introduction mentions SDG 2, it could be more explicit about how the findings of this study will contribute to achieving these goals.

Response 6: We would like to thank you for your deep comment. We appreciated your efforts in reviewing the content. We would like to apologize for we confused you in writing. We had revised and corrected it. As you know our objective is to determine the spatial distribution of inadequate and multilevel factors of MDD and its consequence (the relationship between inadequate MDD and malnutrition among children in SSA which directly tied with SDG 2.2 to end all types of Malnutrition by 2030.

Comment 7: The introduction uses various statistics to highlight the prevalence of malnutrition and inadequate MDD. Ensure that all statistics are up-to-date and properly referenced.

Response 7: We would like to thank you for your deep comment. We appreciated your efforts in reviewing the content. We would like to apologize for we confused you in writing. We had revised and corrected it.

Comment 8: Ensure all terms and abbreviations are clearly defined upon first use. For instance, while MDD is defined, terms like ANC (antenatal care) and PNC (postnatal care) are not immediately explained. A brief definition of these terms when first mentioned would aid in clarity.

Response 8: We would like to thank you for your deep comment. We appreciated your efforts in reviewing the content. We would like to apologize for we confused you in writing. We had revised and corrected it.

Comment 9: The introduction should aim to engage the reader by highlighting the potential impact of the study. Emphasizing how this research can lead to actionable strategies and policy recommendations will undermine its importance.

Response 9: We would like to thank you for your deep comment. We appreciated your efforts in reviewing the content. We would like to apologize for we confused you in writing. We had revised and corrected it.

Here is the importance of the study: Analyzing regional, national, and continental level data on minimum dietary diversity can offer valuable insights for creating contextually relevant strategies and policies. The evidence of geospatial distribution both within and between countries, along with the multilevel factors of inadequate MDD and its consequence among children aged 6 - 23 months are important to meet the expected SDG 2.2 by 2030 (end all types of malnutrition). It provides space-specific evidence, which enables national and sub-national comparisons, tracks changes over time, and identify the high-risk populations. It informs targeted interventions, guides policy decisions, and monitors progress toward nutrition objectives set by the WHO and SDG in Africa.

Methods

Comment 10: Methods, Line 21: Specify the tools or software used for spatial analysis

Response 10: We would like to thank you for your deep comment. We appreciated your efforts in reviewing the content. We would like to apologize for we confused you in writing. We had revised and corrected it. To execute the spatial analysis, the weighted proportions of outcome variables (MDD) with cluster number were computed in STATA version 17 and the spatial analysis was computed using QGIS version 3.28.15, SAT Scan version 9.6, and SAGA GIS version 2.3.2 statistical software.

Comment 11: The manuscript outlines that data were sourced from multiple cross-sectional surveys conducted in SSA in different years. While the nature of surveys are mentioned, there is a lack of detail on how the data from these surveys were harmonized to ensure consistency and comparability. Range of years, information on the survey instruments, sampling methods, and the measures taken to address potential biases in the data collection process would be beneficial. Additionally, details on the specific demographic and geographic variables collected should be provided to understand the data's scope fully.

Response 11: We would like to thank you for your deep comment. We appreciated your efforts in reviewing the content. We would like to apologize for we confused you in writing. We had revised and corrected it. In the current study, we used the kids records (KR) data set and the dependent and independent variables were extracted at https://dhsprogram.com/ by contacting them through personal accounts after justifying the reason for requesting the data (28). The study utilized Demographic and Health Survey (DHS) data from 33 Sub-Saharan African (SSA) countries available from 2010-2020(the detailed information on the specific DHS survey years and countries include are provided in supplementary material).

Regarding geographic variables, geographic information system (GIS) coordinates, were consolidated into a single dataset and the weighted proportions of outcome variables (MDD) with cluster number were computed in STATA version 17 and the spatial analysis was computed using QGIS version 3.28.15, SAT Scan version 9.6, and SAGA GIS version 2.3.2 statistical software.

Comment 12: The study employs a spatial scan statistic to analyze the geographical distribution of your outcome variable in SSA. The chosen methodology allows for the identification of spatial clusters, which is appropriate for the study's aim of detecting geographical patterns. However, additional clarity on the rationale behind selecting this specific statistical method over others would enhance the robustness of the justification. For instance, a comparison with other clustering methods like Bayesian approaches could provide a more comprehensive background for this study.

Response 12: We would like to thank you for your deep comment. We appreciated your efforts in reviewing the content. We would like to apologize for we confused you in writing. Even though, the choice of spatial analysis (spatial stascan) and other methodology like Bayesian approaches depends on the specific problem and available data. In this case, spatial scan statistic provides a more straightforward and efficient solution for predicting values at observed locations by identify clusters, trends, and spatial relationships which is valuable for pattern discovery and decision-making and informing decision support for resource allocation, and environmental management, while Bayesian approaches are valuable for more complex modeling tasks by incorporating prior knowledge, providing uncertainty intervals and can handle non-normal data and complex dependencies.

Comment 13: The discussion rightly connects the high prevalence of inadequate MDD to economic factors and healthcare service disparities. However, it would be beneficial to provide more detailed comparisons with specific regions outside SSA to highlight unique challenges or similarities. This can help in understanding whether similar interventions might be applicable or if entirely new strategies are needed.

Response 13: We would like to thank you for your deep comment. We appreciated your efforts in reviewing the content. We would like to apologize for we confused you in writing. We had revised and corrected it. We state that the prevalence of inadequate MDD is higher than other study finding previous study i

---

## [Decision Letter · Decision Letter 1]

5 Nov 2024

PONE-D-24-12330R1Geospatial Distribution, Multilevel Factors of Inadequate Minimum Dietary Diversity, and its Consequence among Children aged 6-23 months in Sub-Saharan Africa:PLOS ONE

Dear Dr. Endawkie,

Thank you for submitting your manuscript to PLOS ONE. After careful consideration, we feel that it has merit but does not fully meet PLOS ONE’s publication criteria as it currently stands. Therefore, we invite you to submit a revised version of the manuscript that addresses the points raised during the review process.

We look forward to receiving your revised manuscript.

Kind regards,

Jay Saha

Academic Editor

PLOS ONE

Reviewers' comments:

Reviewer's Responses to Questions

**Comments to the Author**

1. If the authors have adequately addressed your comments raised in a previous round of review and you feel that this manuscript is now acceptable for publication, you may indicate that here to bypass the “Comments to the Author” section, enter your conflict of interest statement in the “Confidential to Editor” section, and submit your "Accept" recommendation.

Reviewer #1: All comments have been addressed

2. Is the manuscript technically sound, and do the data support the conclusions?

Reviewer #1: Yes

3. Has the statistical analysis been performed appropriately and rigorously? 

Reviewer #1: Yes

4. Have the authors made all data underlying the findings in their manuscript fully available?

Reviewer #1: Yes

5. Is the manuscript presented in an intelligible fashion and written in standard English?

Reviewer #1: Yes

6. Review Comments to the Author

**Reviewer #1:**  I have seen and reviewed the revised manuscript and carefully considered the authors' responses to the comments and suggestions provided during the review. I am pleased to inform you that the authors have adequately addressed all the concerns raised.

7. PLOS authors have the option to publish the peer review history of their article (what does this mean? ). If published, this will include your full peer review and any attached files.

**Do you want your identity to be public for this peer review?** For information about this choice, including consent withdrawal, please see our Privacy Policy .

Reviewer #1: No

---

## [Author Response · Author response to Decision Letter 2]

6 Nov 2024

Authors’ Responses to Editor and Reviewers Comments

Dear Academic Editor of PLOS One

“Geospatial Distribution and Multilevel Determinants of Inadequate Minimum Dietary Diversity and Its Consequences for Children Aged 6-23 Months in Sub-Saharan Africa"

Dear PLOS One Editors and Reviewers;

We are thankful for your constructive comments. We have looked at the comments and have revised our paper accordingly. We hope our paper improved as a result of incorporating the reviewer’s and academic editor’s comments and suggestions. Here are the authors’ responses to the comments.

Please find for your kind consideration the following:

1. A revised manuscript without track changes.

2. A revised paper with tracked changes

3. A rebuttal letter that responds to each point raised by the academic editor and reviewer.

The point-by-point responses of authors are written by hoping these changes would meet with your favorable consideration, we are happy to hear if there are more comments and suggestions. Please do not hesitate to let us know if you have any questions.

Yours Sincerely

Mr. Abel Endawkie Correspondence Author

Department of Epidemiology and Biostatistics School of Public Health College of Medicine and Health Science Wollo University Dessie Ethiopia

Tel. 251935459310 Email address abelendawkie@gmail.com

We have tried our best to improve it accordingly:

Please revise the manuscript.

Point-by-point response of Authors for editor's and reviewers' comment

Editor's comments to the Authors

Response: We thank you and appreciate your prompt and timely response. To ensure compliance with PLOS ONE's formatting guidelines, we have meticulously followed the typesetting requirements for references, tables, and technical specifications for figures and to expedite the process and prevent any delays.

Reviewer comment for Authors

Reviewer's Responses to previous revision and Questions

Reviewer response

Comment 1: All comments have been addressed

Response 1: Dear reviewer, thank you so much for your kind words about our work! We truly appreciate your thoughtful feedback. Your insights have been invaluable, and we look forward to your thoughts on the changes. Thank you again for your encouraging and positive feedback.

Comment 2: I have seen and reviewed the revised manuscript and carefully considered the authors' responses to the comments and suggestions provided during the review. I am pleased to inform you that the authors have adequately addressed all the concerns raised.

Response 2: Thank you again for your encouraging and positive feedback.

General response to academic editor and editorial office

Dear Editor, we sincerely appreciate your support and encouragement as we prepare for the scientific editing and review of our manuscript titled “Geospatial Distribution and Multilevel Determinants of Inadequate Minimum Dietary Diversity and Its Consequences for Children Aged 6-23 Months in Sub-Saharan Africa.” In this second round of revisions, we would like to clarify that the comments included in the submission folder under the “action link ‘View Attachments’” are those from Reviewer 1 pertaining to the previous review of first round. We have made every effort to address these comments thoroughly and enhance the manuscript to ensure its scientific rigor. Thank you once again for your guidance. We proofread our manuscript to enhance the grammatical and spelling error.

Here is the response below

Comment 1: The research topic: "Spatial Distribution, Multilevel Factors of Inadequate Minimum Dietary Diversity, and its Consequence among Children aged 6-23 months in Sub-Saharan Africa," aligns well with the findings of the manuscript. The study effectively addresses the spatial distribution and multilevel factors, and it also you tried to identify significant health consequences of inadequate dietary diversity. However, the consequences are not as elaborately discussed as the other aspects, which might lead to a perception that the focus was more on factors and distribution. If the consequences were intended to be a primary focus, they could be emphasized more in the discussion and conclusion sections to better align with the stated topic. Otherwise I suggest revision of the topic.

Response 1: We sincerely appreciate your insightful comment and suggestion and we apologize for any confusion in our writing. Our study focuses on the Spatial Distribution, Multilevel Factors of Inadequate Minimum Dietary Diversity, and for policy implication we want to highlight the consequence of inadequate minimum dietary diversity among children 6-23 months age. We indicated that in discussion part: Children who did not consume adequate MDD, anemia, stunting, and wasting are significantly associated. This is finding is supported by other study findings (73-75). This may be because children, who do not consume adequate MDD, it suggests that they are not receiving a sufficient variety of nutrients essential for growth and development. Inadequate MDD can contribute to iron deficiency anemia, wasting and stunting in children due to insufficient intake of essential nutrients needed for proper growth and maintenance of body tissues (76).

Conclusion: This study indicates that in children who did not consume adequate MDD, anemia, stunting, and wasting are significantly associated. Therefore, improving adequate MDD working at determinants of MDD among children helps to achieve the SDG target 2.2 to end all forms of malnutrition by 2030.

Abstract

Comment 2: Abstract, Line 15-16: The phrase "Because of the evidence of geospatial distribution and multilevel factors of inadequate minimum dietary diversity (MDD) and its consequence among children in Sub-Saharan Africa (SSA) remains limited and inadequate feeding practices can lead to malnutrition in young children" need to be rephrased for clarity.

Response 2: We would like to thank you for your deep comment. We appreciated your efforts in reviewing the content. We would like to apologize for we confused you in writing. We had revised and corrected it.

Introduction

Comment 3: The introduction is comprehensive but somewhat dense. While it covers the necessary background, the flow can be improved to enhance readability. Breaking down long sentences

Response 3: We would like to thank you for your deep comment. We appreciated your efforts in reviewing the content. We would like to apologize for we confused you in writing. We had revised and corrected it.

Comment 4: For instance, the first sentence could be split into two: "Adequate minimum dietary diversity (MDD) is crucial for the healthy growth and development of children aged 6 to 23 months. Nutrient adequacy of a diet is influenced by food groupings."

Response 4: We would like to thank you for your deep comment. We appreciated your efforts in reviewing the content. We would like to apologize for we confused you in writing. We had revised and corrected it.

Comment 5: The introduction does a good job of explaining why MDD is important and the consequences of inadequate dietary diversity. However, it would benefit from more specific context about why this study is necessary. What gaps in the existing literature does this study address?

Response 5: We would like to thank you for your deep comment. We appreciated your efforts in reviewing the content. We would like to apologize for we confused you in writing. We had revised and corrected it. Here are the existing evidence and the reason for conducted this research.

Existed evidence: SDG 2 focuses on achieving food security and improving nutrition through universal access to safe and nutritious food to achieve SDG target 2.2 to end all forms of malnutrition (14). However, in SSA, the situation remains contrary to these goals, with statistics showing a rise in malnutrition (15).

Despite the recommendation and significant advances in child nutrition, the prevalence of inadequate MDD is significantly high in SSA and ranged from 67% to 94% in SSA.

Different literatures showed that, the women's education (19-21), the husband's educational status (19-22), child age (20, 23), mass media exposure (19-21), maternal age (23, 24), occupational status (25, 26), wealth index (19-21), antenatal care (22, 27), and residence (26) in were found to be associated with MDD among children aged 6 to 23 months in different countries of SSA.

Importance of study: The evidence of geospatial distribution both within and between countries, along with the multilevel factors of inadequate MDD and its consequence among children aged 6 - 23 months are important to meet the expected SDG 2.2 by 2030 (end all types of malnutrition). It provides space specific evidence, which enables national and sub-national comparisons, tracks changes over time, and identify the high-risk populations. It informs targeted interventions, guides policy decisions, and monitors progress toward nutrition objectives set by the WHO and SDG in Africa.

Gap identified: However, as far as the author searching effort, the evidence of geospatial distribution between/within countries, and multilevel factors of inadequate MDD and its consequence among children aged 6-23 months in SSA are remains limited. Therefore, this study aimed to identify geospatial distribution and multilevel factors of inadequate MDD and its consequence among children aged 6 to 23 months in SSA using recent DHS data.

Comment 6: The connection to the SDGs is a strong point but needs to be more explicitly tied to the study's objectives. While the introduction mentions SDG 2, it could be more explicit about how the findings of this study will contribute to achieving these goals.

Response 6: We would like to thank you for your deep comment. We appreciated your efforts in reviewing the content. We would like to apologize for we confused you in writing. We had revised and corrected it. As you know our objective is to determine the spatial distribution of inadequate and multilevel factors of MDD and its consequence (the relationship between inadequate MDD and malnutrition among children in SSA which directly tied with SDG 2.2 to end all types of Malnutrition by 2030.

Comment 7: The introduction uses various statistics to highlight the prevalence of malnutrition and inadequate MDD. Ensure that all statistics are up-to-date and properly referenced.

Response 7: We would like to thank you for your deep comment. We appreciated your efforts in reviewing the content. We would like to apologize for we confused you in writing. We had revised and corrected it.

Comment 8: Ensure all terms and abbreviations are clearly defined upon first use. For instance, while MDD is defined, terms like ANC (antenatal care) and PNC (postnatal care) are not immediately explained. A brief definition of these terms when first mentioned would aid in clarity.

Response 8: We would like to thank you for your deep comment. We appreciated your efforts in reviewing the content. We would like to apologize for we confused you in writing. We had revised and corrected it.

Comment 9: The introduction should aim to engage the reader by highlighting the potential impact of the study. Emphasizing how this research can lead to actionable strategies and policy recommendations will undermine its importance.

Response 9: We would like to thank you for your deep comment. We appreciated your efforts in reviewing the content. We would like to apologize for we confused you in writing. We had revised and corrected it.

Here is the importance of the study: Analyzing regional, national, and continental level data on minimum dietary diversity can offer valuable insights for creating contextually relevant strategies and policies. The evidence of geospatial distribution both within and between countries, along with the multilevel factors of inadequate MDD and its consequence among children aged 6 - 23 months are important to meet the expected SDG 2.2 by 2030 (end all types of malnutrition). It provides space-specific evidence, which enables national and sub-national comparisons, tracks changes over time, and identify the high-risk populations. It informs targeted interventions, guides policy decisions, and monitors progress toward nutrition objectives set by the WHO and SDG in Africa.

Methods

Comment 10: Methods, Line 21: Specify the tools or software used for spatial analysis

Response 10: We would like to thank you for your deep comment. We appreciated your efforts in reviewing the content. We would like to apologize for we confused you in writing. We had revised and corrected it. To execute the spatial analysis, the weighted proportions of outcome variables (MDD) with cluster number were computed in STATA version 17 and the spatial analysis was computed using QGIS version 3.28.15, SAT Scan version 9.6, and SAGA GIS version 2.3.2 statistical software.

Comment 11: The manuscript outlines that data were sourced from multiple cross-sectional surveys conducted in SSA in different years. While the nature of surveys are mentioned, there is a lack of detail on how the data from these surveys were harmonized to ensure consistency and comparability. Range of years, information on the survey instruments, sampling methods, and the measures taken to address potential biases in the data collection process would be beneficial. Additionally, details on the specific demographic and geographic variables collected should be provided to understand the data's scope fully.

Response 11: We would like to thank you for your deep comment. We appreciated your efforts in reviewing the content. We would like to apologize for we confused you in writing. We had revised and corrected it. In the current study, we used the kids records (KR) data set and the dependent and independent variables were extracted at https://dhsprogram.com/ by contacting them through personal accounts after justifying the reason for requesting the data (28). The study utilized Demographic and Health Survey (DHS) data from 33 Sub-Saharan African (SSA) countries available from 2010-2020(the detailed information on the specific DHS survey years and countries include are provided in supplementary material).

Regarding geographic variables, geographic information system (GIS) coordinates, were consolidated into a single dataset and the weighted proportions of outcome variables (MDD) with cluster number were computed in STATA version 17 and the spatial analysis was computed using QGIS version 3.28.15, SAT Scan version 9.6, and SAGA GIS version 2.3.2 statistical software.

Comment 12: The study employs a spatial scan statistic to analyze the geographical distribution of your outcome variable in SSA. The chosen methodology allows for the identification of spatial clusters, which is appropriate for the study's aim of detecting geographical patterns. However, additional clarity on the rationale behind selecting this specific statistical method over others would enhance the robustness of the justification. For instance, a comparison with other clustering methods like Bayesian approaches could provide a more comprehensive background for this study.

Response 12: We would like to thank you for your deep comment. We appreciated your efforts in reviewing the content. We would like to apologize for we confused you in writing. Even though, the choice of spatial analysis (spatial stascan) and other methodology like Bayesian approaches depends on the specific problem and available data. In this case, spatial scan statistic provides a more straightforward and efficient solution for predicting values at observed locations by identify clusters, trends, and spatial relationships which is valuable for pattern discovery and decision-making and informing decision support for resource allocation, and environmental management, while Bayesian approaches are valuable for more complex modeling tasks by incorporating prior knowledge, providing uncertainty intervals and can handle non-normal data and complex dependencies.

Comment 13: The discussion rightly connects the high prevalence of inadequate

---

## [Decision Letter · Decision Letter 2]

19 Jan 2025

PONE-D-24-12330R2Geospatial Distribution and Multilevel Determinants of Inadequate Minimum Dietary Diversity and Its Consequences for Children Aged 6-23 Months in Sub-Saharan AfricaPLOS ONE

Dear Dr. Endawkie,

Thank you for submitting your manuscript to PLOS ONE. After careful consideration, we feel that it has merit but does not fully meet PLOS ONE’s publication criteria as it currently stands. Therefore, we invite you to submit a revised version of the manuscript that addresses the points raised during the review process.

We look forward to receiving your revised manuscript.

Kind regards,

Dev Ram Sunuwar, MS

Academic Editor

PLOS ONE

Journal Requirements:

Reviewers' comments:

Reviewer's Responses to Questions

**Comments to the Author**

1. If the authors have adequately addressed your comments raised in a previous round of review and you feel that this manuscript is now acceptable for publication, you may indicate that here to bypass the “Comments to the Author” section, enter your conflict of interest statement in the “Confidential to Editor” section, and submit your "Accept" recommendation.

Reviewer #2: All comments have been addressed

2. Is the manuscript technically sound, and do the data support the conclusions?

Reviewer #2: Yes

3. Has the statistical analysis been performed appropriately and rigorously? 

Reviewer #2: Yes

4. Have the authors made all data underlying the findings in their manuscript fully available?

Reviewer #2: Yes

5. Is the manuscript presented in an intelligible fashion and written in standard English?

Reviewer #2: Yes

6. Review Comments to the Author

Reviewer #2: The paper calculates sub-national estimates for Minimum Dietary Diversity (MMD) for children aged 6-23 months across 33 out of 40 countries in Sub-Saharan Africa using data from the Demographic and Health Surveys from 2010-20. In addition to identifying and mapping these descriptive statistics on an important variable, the authors also seek to identify covariates associated with variations in MDD. This study stands out in that it provides a harmonized methodology for comparison on MDD across such a wide variety of countries in the region, and does so at a subnational level. It is also important in the way it draws attention to the associations between MDD and predictors of undernutrition such as stunting and wasting, thus providing a window to policy solutions rather than just highlighting the problems.

I have only one minor suggestion, one minor correction to comment on and the remainder are points for discussion on further work. I highly recommend the work for publication, especially seeing how the comments from the previous reviewer have been diligently addressed.

The first minor comment would be that in L81-83 of the motivation, they state that existing studies of this nature 'remain limited', it would be useful if the authors could find some of the closest equivalents and highlight where they have not gone as far as this one does.

Secondly, I think there may be a minor error in the phrasing of the interpretation of the result regarding distance to health facility. In line 295, it says 'longer distance to health facilities was associated with lower odds of inadequate MDD' when the table shows the opposite result - that those who stated that distance to a health facility was 'not a big problem' were less likely to have inadequate MDD. I would suggest you correct this. It seems to have been correctly interpreted in the discussion, so this must have been a typo in an earlier version.

Regarding the results, they seem to be extremely interesting. For example, the detection of the association between inadequate MDD and lower media exposure is helpful from a policy perspective when thinking about encouraging behavioural change if food sources to increase dietary diversity are already available. It would be interesting as part of future work to look at the extent to which MDD is more attributable to issues of availability (e.g. crop growth, market connectivity) versus cultural/social norms. If such factors were added into the analysis that is presented, it could help to determine to what extent it is easier or more difficult for populations to improve their dietary diversity with simple information campaigns versus market expansion/changes to agricultural practices i.e. to what extent MDD inadequacy is a supply versus a demand problem. Global satellite imagery is increasingly available and could be used to examine the association between cropping patterns and MDD (https://earth.esa.int/eogateway/success-story/predicting-crop-yield-using-planet-data). It would also be very interesting to incorporate global satellite data on friction (ease/difficulty of travel) or distance to road networks (https://data.malariaatlas.org/). Including such additional factors would help to prioritise what kinds of supply side constraints could be better prioritised.

Finally, one other limitation in the results that I would highlight is that though the relationships that are identified are important, they are also only weakly significant in each case. It is mentioned that data is used across a 10 year period. Perhaps in future work, it may be helpful to refine the analysis to a cross section within a more limited time range so as to reduce noise and allow relationships to come through more strongly. In that sense, I think the date ranges for the analysis should be added to Table 3 for each of the countries. That would be my final minor suggestion.

7. PLOS authors have the option to publish the peer review history of their article (what does this mean? ). If published, this will include your full peer review and any attached files.

**Do you want your identity to be public for this peer review?** For information about this choice, including consent withdrawal, please see our Privacy Policy .

Reviewer #2: **Yes: ** Sophie C E Ayling

---

## [Author Response · Author response to Decision Letter 3]

22 Jan 2025

Authors’ Responses to Editor and Reviewers Comments

Dear Academic Editor of PLOS One

“Geospatial Distribution and Multilevel Determinants of Inadequate Minimum Dietary Diversity and Its Consequences for Children Aged 6-23 Months in Sub-Saharan Africa"

Dear PLOS One Editors and Reviewers;

We are thankful for your constructive comments. We have looked at the comments and have revised our paper accordingly. We hope our paper improved as a result of incorporating the reviewer’s and academic editor’s comments and suggestions. Here are the authors’ responses to the comments.

Please find for your kind consideration the following:

1. A revised manuscript without track changes.

2. A revised paper with tracked changes

3. A rebuttal letter that responds to each point raised by the academic editor and reviewer.

The point-by-point responses of authors are written by hoping these changes would meet with your favorable consideration, we are happy to hear if there are more comments and suggestions. Please do not hesitate to let us know if you have any questions.

Yours Sincerely

Mr. Abel Endawkie Correspondence Author

Department of Epidemiology and Biostatistics School of Public Health College of Medicine and Health Science Wollo University Dessie Ethiopia

Tel. 251935459310 Email address abelendawkie@gmail.com

We have tried our best to improve it accordingly:

Please revise the manuscript.

Point-by-point response of Authors for editor's and reviewers' comment

Editor's comments to the Authors

Response: We thank you and appreciate your prompt and timely response. To ensure compliance with PLOS ONE's formatting guidelines, we have meticulously followed the typesetting requirements for references, tables, and technical specifications for figures and to expedite the process and prevent any delays.

Reviewer comment for Authors

Reviewer #2: The paper calculates sub-national estimates for Minimum Dietary Diversity (MMD) for children aged 6-23 months across 33 out of 40 countries in Sub-Saharan Africa using data from the Demographic and Health Surveys from 2010-20. In addition to identifying and mapping these descriptive statistics on an important variable, the authors also seek to identify covariates associated with variations in MDD. This study stands out in that it provides a harmonized methodology for comparison on MDD across such a wide variety of countries in the region, and does so at a subnational level. It is also important in the way it draws attention to the associations between MDD and predictors of undernutrition such as stunting and wasting, thus providing a window to policy solutions rather than just highlighting the problems.

Response 1: Thank you for your positive and encouraging feedback on our study! We greatly appreciate your thoughtful insights and are glad to hear that you found our methodology and findings valuable. Your comments on the significance of our work in highlighting the associations between Minimum Dietary Diversity (MDD) and undernutrition are particularly encouraging. We look forward to your further thoughts on our revisions. Thank you once again for your support!

Comment 1: I have only one minor suggestion, one minor correction to comment on and the remainder are points for discussion on further work. I highly recommend the work for publication, especially seeing how the comments from the previous reviewer have been diligently addressed.

Comment 2: The first minor comment would be that in L81-83 of the motivation, they state that existing studies of this nature 'remain limited', it would be useful if the authors could find some of the closest equivalents and highlight where they have not gone as far as this one does.

Response 2: Thank you for your valuable feedback on our manuscript! We appreciate your suggestion regarding the need to identify and discuss existing studies in our motivation section. We have made revisions to include relevant studies and highlight how our work extends beyond them. Your input has been instrumental in enhancing the clarity of our research. Thank you once again for your support!

Comment 3: Secondly, I think there may be a minor error in the phrasing of the interpretation of the result regarding distance to health facility. In line 295, it says 'longer distance to health facilities was associated with lower odds of inadequate MDD' when the table shows the opposite result - that those who stated that distance to a health facility was 'not a big problem' were less likely to have inadequate MDD. I would suggest you correct this. It seems to have been correctly interpreted in the discussion, so this must have been a typo in an earlier version.

Response 3: Dear Reviewer, thank you for your thoughtful feedback and kind words about our work! We sincerely apologize for the misinterpretation, which does not align with the actual findings. We have revised the results section accordingly based on your recommendation. Thank you once again for your encouraging and constructive comments!

Comment 4: Regarding the results, they seem to be extremely interesting. For example, the detection of the association between inadequate MDD and lower media exposure is helpful from a policy perspective when thinking about encouraging behavioural change if food sources to increase dietary diversity are already available. It would be interesting as part of future work to look at the extent to which MDD is more attributable to issues of availability (e.g. crop growth, market connectivity) versus cultural/social norms. If such factors were added into the analysis that is presented, it could help to determine to what extent it is easier or more difficult for populations to improve their dietary diversity with simple information campaigns versus market expansion/changes to agricultural practices i.e. to what extent MDD inadequacy is a supply versus a demand problem. Global satellite imagery is increasingly available and could be used to examine the association between cropping patterns and MDD (https://earth.esa.int/eogateway/success-story/predicting-crop-yield-using-planet-data). It would also be very interesting to incorporate global satellite data on friction (ease/difficulty of travel) or distance to road networks (https://data.malariaatlas.org/). Including such additional factors would help to priorities what kinds of supply side constraints could be better prioritized.

Response 4: Dear Reviewer, thank you for your encouraging feedback and for highlighting the intriguing aspects of our findings! We appreciate your suggestion to explore the relative contributions of availability and cultural/social norms to inadequate MDD in future research. Incorporating factors such as crop growth, market connectivity, and the use of global satellite imagery could indeed provide a deeper understanding of the complexities surrounding dietary diversity. We agree that analyzing these elements could help differentiate between supply-side and demand-side challenges, ultimately guiding more effective policy interventions. Your suggestions for utilizing satellite data on travel friction and road networks are particularly insightful. We acknowledge the suggestion which is important thought, However the DHS data has no data like crop growth, and market connectivity. Therefore, we will certainly consider these approaches in our future work. Thank you once again for your thoughtful comments!

Comment 5: Finally, one other limitation in the results that I would highlight is that though the relationships that are identified are important, they are also only weakly significant in each case. It is mentioned that data is used across a 10 year period. Perhaps in future work, it may be helpful to refine the analysis to a cross section within a more limited time range so as to reduce noise and allow relationships to come through more strongly. In that sense, I think the date ranges for the analysis should be added to Table 3 for each of the countries. That would be my final minor suggestion.

Response 5: Dear Reviewer, thank you for your thoughtful feedback and kind words about our work! We have revised the results section accordingly based on your recommendation. Thank you once again for your encouraging and constructive comments!

---

## [Decision Letter · Decision Letter 3]

16 Feb 2025

PONE-D-24-12330R3Geospatial Distribution and Multilevel Determinants of Inadequate Minimum Dietary Diversity and Its Consequences for Children Aged 6-23 Months in Sub-Saharan AfricaPLOS ONE

Dear Dr. Hailu,

Thank you for submitting your manuscript to PLOS ONE. After careful consideration, we feel that it has merit but does not fully meet PLOS ONE’s publication criteria as it currently stands. Therefore, we invite you to submit a revised version of the manuscript that addresses the points raised during the review process.

We look forward to receiving your revised manuscript.

Kind regards,

Dev Ram Sunuwar, MS

Academic Editor

PLOS ONE

Journal Requirements:

Reviewers' comments:

Reviewer's Responses to Questions

**Comments to the Author**

1. If the authors have adequately addressed your comments raised in a previous round of review and you feel that this manuscript is now acceptable for publication, you may indicate that here to bypass the “Comments to the Author” section, enter your conflict of interest statement in the “Confidential to Editor” section, and submit your "Accept" recommendation.

Reviewer #2: (No Response)

2. Is the manuscript technically sound, and do the data support the conclusions?

Reviewer #2: Yes

3. Has the statistical analysis been performed appropriately and rigorously? 

Reviewer #2: Yes

4. Have the authors made all data underlying the findings in their manuscript fully available?

Reviewer #2: Yes

5. Is the manuscript presented in an intelligible fashion and written in standard English?

Reviewer #2: Yes

6. Review Comments to the Author

Reviewer #2: Thank you for addresssing the comments in the response letter. I would have liked to see a bit more added in the discussion about the other opportunities for future work, but otherwise all the more immediate issues seem to be addressed.

One note is that I left in my previous revision when asked for 'any other comments' was about the figures. I didn't leave those observations in the main body of the comments but they seem to have not reached you. I'm putting them again here:

- Fig 1 - add title

- Fig 2 - rather than stating 'legend' as the title, could you put the name of the variable? e.g. % of child population age XX meeting MDD? Or add a title, with a less detailed legend?

- Fig 3 - likewise, add a title, make legend a bit clearer as to what 'high' 'low' and 'not significant' refer to

- Fig 4 - same comment to provide a title and provide a clearer labelling for the legend. does 100 mean that the location has 100% of the child population meeting MDD?

- Fig 5 - it's not clear if the circles refer to countries or areas within countries. Please also add a title to the plot. Presumably the numbers are showing rankings. The resolution is not super clear, you may want to make sure you save it as a PNG with 300 DPI at least.

- Fig 6 - also needs a title

I think those figure edits will be necessary for publication as many are missing clear legends or titles. Otherwise, congratulations on your work.

7. PLOS authors have the option to publish the peer review history of their article (what does this mean? ). If published, this will include your full peer review and any attached files.

**Do you want your identity to be public for this peer review?** For information about this choice, including consent withdrawal, please see our Privacy Policy .

Reviewer #2: **Yes: ** Sophie Ayling

---

## [Author Response · Author response to Decision Letter 4]

18 Feb 2025

Authors’ Responses to Editor and Reviewers Comments

Dear Academic Editor of PLOS One

“Geospatial Distribution and Multilevel Determinants of Inadequate Minimum Dietary Diversity and Its Consequences for Children Aged 6-23 Months in Sub-Saharan Africa"

Dear PLOS One Editors and Reviewers;

We are thankful for your constructive comments. We have looked at the comments and have revised our paper accordingly. We hope our paper improved as a result of incorporating the reviewer’s and academic editor’s comments and suggestions. Here are the authors’ responses to the comments.

Please find for your kind consideration the following:

1. A revised manuscript without track changes.

2. A revised paper with tracked changes

3. A rebuttal letter that responds to each point raised by the academic editor and reviewer.

The point-by-point responses of authors are written by hoping these changes would meet with your favorable consideration, we are happy to hear if there are more comments and suggestions. Please do not hesitate to let us know if you have any questions.

Yours Sincerely

Mr. Abel Endawkie Correspondence Author

Department of Epidemiology and Biostatistics School of Public Health College of Medicine and Health Science Wollo University Dessie Ethiopia

Tel. 251935459310 Email address abelendawkie@gmail.com

We have tried our best to improve it accordingly:

Please revise the manuscript.

Point-by-point response of Authors for editor's and reviewers' comment

Reviewer comment for Authors

Reviewer #2: Thank you for addressing the comments in the response letter. I would have liked to see a bit more added in the discussion about the other opportunities for future work, but otherwise all the more immediate issues seem to be addressed.

Response 1: Thank you for your positive and encouraging feedback on our study! We greatly appreciate your thoughtful insights and are glad to hear that you found our methodology and findings valuable.

Comment 2: One note is that I left in my previous revision when asked for 'any other comments' was about the figures. I didn't leave those observations in the main body of the comments but they seem to have not reached you. I'm putting them again here:

- Fig 1 - add title

- Fig 2 - rather than stating 'legend' as the title, could you put the name of the variable? e.g. % of child population age XX meeting MDD? Or add a title, with a less detailed legend?

- Fig 3 - likewise, add a title, make legend a bit clearer as to what 'high' 'low' and 'not significant' refer to

- Fig 4 - same comment to provide a title and provide a clearer labelling for the legend. does 100 mean that the location has 100% of the child population meeting MDD?

- Fig 5 - it's not clear if the circles refer to countries or areas within countries. Please also add a title to the plot. Presumably the numbers are showing rankings. The resolution is not super clear, you may want to make sure you save it as a PNG with 300 DPI at least.

- Fig 6 - also needs a title. I think those figure edits will be necessary for publication as many are missing clear legends or titles. Otherwise, congratulations on your work

Response: Dear Reviewer, Thank you for your thoughtful feedback and kind words about our work! I appreciate your observations regarding the figures, and I apologize for any confusion in previous communications. We have revised and provided the title in the main document as outlined by the journal guideline. We have made the necessary revisions based on your recommendations. Specifically, we have added titles to all figures. We believe these edits will enhance the clarity of the figures and are necessary for publication. Thank you once again for your encouraging and constructive comments!

---

## [Editor Report · Decision Letter 4]

4 Mar 2025

Geospatial Distribution and Multilevel Determinants of Inadequate Minimum Dietary Diversity and Its Consequences for Children Aged 6-23 Months in Sub-Saharan Africa

PONE-D-24-12330R4

Dear Dr. Endawkie,

We’re pleased to inform you that your manuscript has been judged scientifically suitable for publication and will be formally accepted for publication once it meets all outstanding technical requirements.

Kind regards,

Dev Ram Sunuwar, MS

Academic Editor

PLOS ONE
---

## [Editor Report · Acceptance letter]

PONE-D-24-12330R4

PLOS ONE

Dear Dr. Endawkie,

I'm pleased to inform you that your manuscript has been deemed suitable for publication in PLOS ONE. Congratulations! Your manuscript is now being handed over to our production team.

Kind regards,

on behalf of

Mr Dev Ram Sunuwar

Academic Editor

PLOS ONE